



# Emulating Earth System Model temperatures: from global mean temperature trajectories to grid-point level realizations on land

Lea Beusch[1], Lukas Gudmundsson[1], and Sonia I. Seneviratne[1]

[1]Institute for Atmospheric and Climate Science, ETH Zurich, Zurich, Switzerland

**Correspondence:** Lea Beusch (lea.beusch@env.ethz.ch)

**Abstract.** Earth System Models (ESMs) are invaluable tools to study the climate system's response to specific greenhouse gas emission pathways. Large single-model initial-condition and multi-model ensembles are used to investigate the range of possible responses and serve as input to climate impact and integrated assessment models. Thereby, climate signal uncertainty is propagated along the uncertainty chain and its effect on interactions between humans and the Earth system can be quantified. However, generating both single-model initial-condition and multi-model ensembles is computationally expensive. In this study, we assess the feasibility of geographically-explicit climate model emulation, i.e., of statistically producing large ensembles of global spatially and temporally correlated land temperature field time series at a negligible computational cost which closely resemble ESM runs spanning from 1870 to 2099. For this purpose, we develop a modular framework that consists of (i) a global mean temperature emulator, (ii) a local mean temperature emulator, and (iii) a local residual temperature variability emulator. We first show that to successfully mimic single-model initial-condition ensembles of yearly temperature, it is sufficient to train on a single ESM run, but separate emulators need to be calibrated for individual ESMs given fundamental inter-model differences. We then emulate 40 climate models of the Coupled Model Intercomparison Project Phase 5 (CMIP5) to create a "super-ensemble", i.e., a large ensemble that closely resembles a multi-model initial-condition ensemble. Furthermore, the thereby emerging ESM-specific emulator calibration parameters provide essential insights on inter-model divergences across a broad range of scales which can be viewed as a "model ID" of core properties of each ESM. Our results highlight that, for temperature at the spatio-temporal scales considered here, it is likely more advantageous to invest computational resources into generating multi-model ensembles rather than large single-model initial-condition ensembles. Such multi-model ensembles can then be extended to super-ensembles with geographically-explicit temperature emulators like the one presented here.

# 1 Introduction

The range of climate responses to external radiative forcing is affected by both internal variability and inter-model differences (Hawkins and Sutton, 2009; Deser et al., 2012; Taylor et al., 2012). While inter-model uncertainty is typically accounted for by





considering simulations from several climate models (Meehl et al., 2007; Taylor et al., 2012; Eyring et al., 2016), uncertainty
due to internal climate variability is usually quantified through running the same climate model a number of times with slightly
different initial conditions (Deser et al., 2012; Fischer et al., 2013; Kay et al., 2015; Leduc et al., 2019; Maher et al., 2019).

As climate model ensembles are inherently expensive to run, there is an interest in approximating Earth System Model
(ESM) output by computationally cheap emulators. In the field of climate science, the term emulator is used for a variety
of statistical models which learn from existing runs of complex climate models to infer properties of runs which have not
been generated yet. This allows for an exploration of the emulated space at a lower computational cost. ESM emulators target
different aspects of the climate system. For example, some emulators focus on the impacts of sub-grid scale parameterizations
(Rougier et al., 2009; Williamson et al., 2013). Others target the effect of greenhouse gas emission scenarios on global mean
temperature (Meinshausen et al., 2011; Goodwin, 2016) or on regional mean climate variable fields (Santer et al., 1990; Tebaldi
and Arblaster, 2014; Tebaldi and Knutti, 2018). Furthermore, there are emulators focusing on regional-scale internal variability
(Castruccio and Genton, 2016; Alexeeff et al., 2018; Link et al., 2019). Recently, first attempts have additionally been made to
emulate the full dynamics of simple general circulation models (Scher, 2018; Scher and Messori, 2019). In this study, the term
emulator is used to refer to a computationally cheap statistical tool which generates additional realizations of geographically-
explicit and spatio-temporally correlated land temperature field time series for a specific greenhouse gas emission pathway
at a yearly resolution. The emulator thus produces realizations which closely resemble initial-condition ensemble members
of the considered ESMs. In the context of large multi-model ensembles, the computationally cheap emulator can be used
to produce look-alikes of large initial-condition ensembles for every model within the multi-model ensemble resulting in a
"super-ensemble", i.e., a large ensemble which closely resembles a multi-model initial-condition ensemble.

To build this statistical temperature emulator, an overarching modular emulation framework is proposed and put into context
of previous work in Sect. 2. The employed data is described in Sect. 3, and the specific implementation of the proposed
framework is introduced in Sect. 4. To visualize the characteristics and capabilities of the emulator, calibration results and
example realizations are shown for four selected ESMs in Sect 5. In Sect. 6, the emulator is applied to a large multi-model
ensemble containing 40 climate models. In addition to calibration results and example realizations, quantitative verification
of in-sample and out-of-sample performance are presented. In Sect. 7, the results are discussed and finally, in Sect. 8, the
conclusions and an outlook are provided.

## 2   A framework for end-to-end climate model emulation

We propose an additive framework for temperature emulation at the yearly scale for a specific greenhouse gas emission pathway
which can be summarized as

$$T_{s,t} = f(T_t^{glob}) + \eta_{s,t}, \tag{1}$$

where the temperature $T$ at grid point $s$ and time $t$ is defined as a deterministic response to the current global mean temperature
$T_t^{glob}$, indicated by the function $f()$, and a stochastic local residual temperature variability term $\eta_{s,t}$. Deterministic contribu-
tions from physical feedbacks other than the ones captured within the global mean temperature signal are thus neglected. The





assumption of an underlying additivity is in line with frequently employed approaches in climate science uncertainty analysis (Hawkins and Sutton, 2009) and climate change detection and attribution studies (Allen and Stott, 2003).

Our framework requires three sub-modules. An emulator for global mean temperature, an emulator for the deterministic mean grid-point level response to the global temperature, and a local residual temperature variability emulator. In the following, we place existing literature within these sub-modules. As this study is primarily concerned with temperature, we focus solely on this variable in our literature review. However, several of the referred studies treat additional variables such as precipitation (e.g., Tebaldi and Arblaster, 2014; Seneviratne et al., 2016; Wartenburger et al., 2017) or cloud cover (e.g., Osborn et al., 2016).

*Global mean temperature emulators*

Emulated global mean temperature is often an output of computationally efficient simple energy-balance climate models (Meinshausen et al., 2011; Goodwin, 2016). While such models provide an estimate of the forced global temperature response, they cannot quantify the effects of interannual climate variability. Statistical models accounting for temporal autocorrelation, however, can be used to generate stochastic ensembles of unforced interannual variability of global mean temperature (Brown et al.,
70   2015).

*Local mean temperature emulators*

Pattern scaling is a frequently employed approach to relate grid-point level temperature to global mean temperature and is also used to emulate warming patterns across emission scenarios (Santer et al., 1990; Mitchell, 2003; Tebaldi and Arblaster,
2014). It was originally introduced by Santer et al. (1990) and different implementations exist (Mitchell, 2003). Most often, the temperature pattern provided by a climate model is averaged over a late 21$^{st}$ century multi-decadal time period and the associated average global mean temperature is obtained (Tebaldi and Arblaster, 2014). This pattern is then linearly interpolated to a desired global mean temperature level. An alternative is to extract the pattern of change from a transient simulation at the time when this simulation reaches the desired global mean temperature on its way to higher warming (Herger et al., 2015;
Seneviratne et al., 2016; King et al., 2017). Other approaches include carrying out a linear regression (Lynch et al., 2017) or fitting a linear mixed-effect model (Alexeeff et al., 2018) to global mean temperature at each grid point individually.

The most important assumption underlying pattern scaling is that local mean temperatures are linearly related to global mean temperature and that this relationship is consistent across forcing scenarios. For surface temperature on land this assumption is satisfactorily met (Mitchell, 2003; Tebaldi and Arblaster, 2014; Seneviratne et al., 2016; Wartenburger et al., 2017; Os-
born et al., 2018). However, for strong mitigation scenarios and under strong aerosol forcing, pattern scaling is less accurate (May, 2012; Levy et al., 2013). An additional assumption of pattern scaling is that external forcing and internal variability are independent which may not always be true and could lead to pattern estimation errors (Lopez et al., 2014).

More complex local mean temperature emulation methods are rare and often directly conditioned on $CO_2$ concentration profiles instead of global mean temperature (Castruccio et al., 2014; Holden and Edwards, 2010). For instance, it has been pro-
posed to employ past trajectories of atmospheric $CO_2$ to model regional temperatures with an infinite distributed lag model to capture non-linear behaviour in spatial patterns for regional scale emulation (Castruccio et al., 2014) as well as within a global





space-time model (Castruccio and Stein, 2013). Other authors use singular value decomposition to emulate decadal temperature fields across scenarios while accounting for complex spatio-temporal feedbacks (Holden and Edwards, 2010; Holden et al., 2014).


*Local residual temperature variability emulators*

Several approaches exists to emulate local residual temperature variability based on observations and climate model simulations (Osborn et al., 2016; McKinnon et al., 2017; Castruccio and Stein, 2013; Link et al., 2019; Alexeeff et al., 2018). Observations can be employed to avoid climate model biases but are limited to rather short observational records when deriving the local

temperature variability properties (Osborn et al., 2016; McKinnon et al., 2017; McKinnon and Deser, 2018). The simplest approach is to detrend observed time series of temperature and obtain additional realization by shifting the starting date of the time series (Osborn et al., 2016). More realizations have been generated from an observational record by resampling spatial fields of detrended observed local temperature variability in blocks of two years at the cost of loosing longer-term memory (McKinnon et al., 2017; McKinnon and Deser, 2018).

When employing ESM temperature time series instead, longer time series as well as multiple climate run realizations are available to derive the statistical properties of the local residual temperature variability. Several authors fit autoregressive (AR) models to a set of climate model runs to account for temporal autocorrelation when simulating additional realizations of local residual temperature variability (Castruccio and Stein, 2013; Castruccio et al., 2014; Castruccio and Genton, 2016; Bao et al., 2016). Most of them furthermore consider the spatial dependence in the innovation terms of the AR models by parameterizing

their covariance structure with a Matérn covariance function (Castruccio and Stein, 2013; Castruccio and Genton, 2016; Bao et al., 2016). An alternative approach is based on decomposing detrended ESM runs into their principal components, whose phases are randomly perturbed to generate additional realizations of local residual temperature variability (Link et al., 2019). All approaches listed so far assume temporal stationarity in the local residual climate variability. Large single-model initial-condition ensembles, however, can be used to re-sample detrended temperature fields within a certain window size around a

global mean temperature level which makes it possible to account for temporal non-stationarities in local residual temperature variability in a warming world (Alexeeff et al., 2018). To enlarge the amount of fields to sample from, a method has additionally been proposed to stochastically simulate spatially non-stationary Gaussian fields with a LatticeKrig model (Nychka et al., 2018).

Overall, most studies rely on the assumption that local residual temperature variability is stationary in time which is known

not to be fulfilled everywhere. Olonscheck and Notz (2017) and references therein provide a comprehensive overview on possible changes in temperature variability in the historical time period and a business-as-usual greenhouse gas emission scenario for models in the large Coupled Model Intercomparison Project Phase 5 (CMIP5) multi-model ensemble. They find that the strongest and most likely changes will occur over oceans but also point out land regions where variability is projected to change in the future. During the historical time period, they identify only weak changes in the variability.


## 3 Data

### 3.1 Data sources

Runs from 40 CMIP5 climate models (Taylor et al., 2012) covering the historical time period (1870–2005) and the business-as-usual greenhouse gas emission scenario RCP8.5 (2005–2099, Riahi et al., 2011) are employed. The models, the associated modeling groups, and the number of initial-condition member used here are listed in Table A1. Out of these 40 models, a special focus is set on four ESMs (CanESM2, CESM1(CAM5), HadGEM2-ESM, MPI-ESM-LR). These are chosen because they represent different model genealogies according to Knutti et al. (2013) and because they provide several initial-condition ensemble members. Additionally, stratospheric aerosol optical depth is used as a proxy for volcanic activity during the historical time period. This aerosol dataset was originally described by Sato et al. (1993) and later updated to cover the considered time period.

### 3.2 Data processing

Here, we focus on surface temperature anomaly at a yearly resolution. Temperature fields were bilinearly interpolated onto a 2.5° x 2.5°grid resulting in 3043 land grid points for each climate model. Yearly mean temperatures were computed at each grid point and the average over the reference period of 1870–1899 at the respective grid points was subtracted. In the text, for simplicity reasons, we use the term "temperature" when referring to "yearly surface temperature anomaly". For the stratospheric aerosol optical depth, the globally averaged yearly time series is employed.

Whenever regional averages are shown, area-weighted means are referred to. The regions employed in this study are 26 SREX land regions (Seneviratne et al., 2012) as well as global mean and global land mean (Fig. 1). While global mean refers to the average across all grid points, global land mean refers to the average across all land grid points excluding Antarctica.

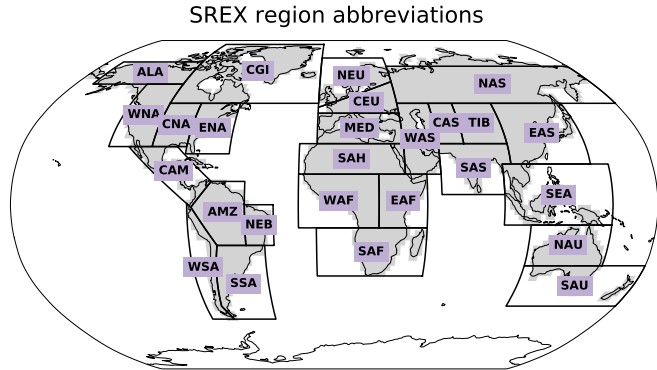

**Figure 1.** Map of the SREX regions and their abbreviations. The considered land grid points are shown in grey.





## 4 Methods

### 4.1 General approach and terminology

### Emulation framework

**Figure 2.** Illustration of the emulator framework. The deterministic modules of the emulator are marked in blue, the stochastic ones in orange.

To emulate temperature fields at the yearly scale for a specific greenhouse gas emission pathway, we follow the framework introduced in Sect. 2. Figure 2 shows the chosen implementation graphically and the associated governing equation reads

$$T_{s,t} = f(T_t^{glob,det}, T_t^{glob,var}) + \eta_{s,t}. \tag{2}$$

Hence, the global forcing is split into a deterministic trend $T_t^{glob,det}$ and a stochastic variability $T_t^{glob,var}$ term, both of which contribute to the local grid-point temperature $T_{s,t}$ in a deterministic way via the function $f()$. The residual spatio-temporally correlated local temperature variability $\eta_{s,t}$ is modeled stochastically.

For each considered climate model, an emulator is trained on a single climate model run spanning 230 years (1870–2099). To calibrate the emulators, the land temperature field time series as well as the global mean temperature trajectory are required.





The emulators' performance is subsequently evaluated with in-sample and, where possible, additionally, out-of-sample ver-
ification. Thereby, the in-sample verification is carried out on the training run itself and indicates how successfully our frame-
work implementation captures the training run. For climate models with several initial-condition ensemble members, out-of-
sample verification is conducted on the runs not employed during the training of the emulators. Hence, the out-of-sample
verification serves as a proxy for the emulators' capability to mimic true ESM initial-condition ensembles.

In the following sections, the chosen equations for each emulator module are introduced and the respective calibration
procedures are described. Afterwards, the emulator performance evaluation approach is explained.

### 4.2    Framework implementation

#### 4.2.1    Global mean temperature emulator

The global mean temperature emulator stochastically creates additional realizations of global mean temperature time series.

Global mean temperature is separated into a deterministic part $T_t^{glob,det}$ shared by all emulations and a stochastic global
variability part $T_t^{glob,var}$ varying between realizations:

$$T_t^{glob} = T_t^{glob,det} + T_t^{glob,var}. \tag{3}$$

In the deterministic trend, $T_t^{glob,det}$, smooth forcing $T_t^{glob,sm}$ and abrupt changes induced by volcanic eruptions $T_t^{glob,volc}$
are accounted for in an additive way:

$$T_t^{glob,det} = T_t^{glob,sm} + T_t^{glob,volc}. \tag{4}$$

First, $T_t^{glob,sm}$ is derived by locally weighted scatterplot smoothing (LOWESS) of $T_t^{glob}$.

In a next step, $T_t^{glob,volc}$ is approximated as the linear response of the residuals of the smooth trend, i.e., $(T_t^{glob} - T_t^{glob,sm})$,
to stratospheric aerosol optical depth $AOD_t$ with regression coefficients $\lambda_0$ and $\lambda_1$:

$$T_t^{glob,volc} = \lambda_0 + \lambda_1 \cdot AOD_t \tag{5}$$

The time series of unforced global temperature variability $T_t^{glob,var} = T_t^{glob} - T_t^{glob,det}$ is modeled as an AR process of
order p with coefficients $\alpha_0,...,\alpha_p$ such that

$$T_t^{glob,var} = \alpha_0 + \sum_{k=1}^{k=p} \alpha_k \cdot T_{t-k}^{glob,var} + \epsilon_t \quad \text{with} \quad \epsilon_t \sim \mathcal{N}(0,\sigma), \tag{6}$$

whereby $\epsilon_t$ is a white noise innovation term drawn from a Gaussian distribution with mean zero and its standard deviation set
to the empirical standard deviation of the innovations of the training samples ($\sigma$).

Here, the LOWESS smoothing window length is 50 years with weights decaying with increasing distance according to a
tricube weight function. The regression coefficients for the forced response to volcanic eruptions are obtained with the ordinary
least squares (OLS) algorithm. The coefficients of the AR process are fit by means of maximum likelihood and the Bayesian
Information Criterion (BIC), which punishes model complexity to avoid overfitting, is employed to select the order p with the
maximum possible order set to 8.





### 4.2.2 Local mean temperature emulator

The local mean temperature emulator translates the global temperature signal into a grid-point level response $T_{s,t}^{resp}$. Due to the pronounced linear scaling of regional land temperatures with global mean temperature (Seneviratne et al., 2016; Wartenburger et al., 2017), the local mean temperature $T_{s,t}^{resp}$ is expressed as

$$T_{s,t}^{resp} = f(T_t^{glob,det}, T_t^{glob,var}) = \beta_s^{det} \cdot T_t^{glob,det} + \beta_s^{var} \cdot T_t^{glob,var} + \beta_s^{int}, \tag{7}$$

with regression coefficients $\beta_s^{det}, \beta_s^{var}$, and $\beta_s^{int}$. Hence, the response of the local mean temperature to the deterministic global trend $T_t^{glob,det}$ and the response to the global temperature variability $T_t^{glob,var}$ are separately taken into account.

In this study, the linear regression coefficients are estimated with OLS at each grid point.

### 4.2.3 Local residual temperature variability emulator

The local residual temperature variability refers to the spatio-temporally correlated residual variability which cannot be accounted for through a response to the global mean temperature signal. This variability is assumed to be stationary in time and Gaussian in nature which makes it possible to model the time series as local AR processes with spatially correlated innovations (Humphrey and Gudmundsson, 2019). Hence, additional realizations of the local residual temperature variability are generated according to

$$\eta_{s,t} = \gamma_{0,s} + \sum_{k=1}^{k=p_s} \gamma_{k,s} \cdot \eta_{s,t-k} + \nu_{s,t} \quad \text{with} \quad \nu_{s,t} \sim \mathcal{N}(0, \Sigma(r)), \tag{8}$$

whereby $\gamma_{0,s},...,\gamma_{p_s,s}$ refer to the coefficients of the AR model and $\nu_{s,t}$ are the white noise innovations drawn from a multivariate Gaussian with mean zero and the regularized empirical spatial covariance matrix $\Sigma(r)$.

The rank deficient empirical covariance matrix $\tilde{\Sigma}$ needs to be regularized to obtain a robust estimate of the co-variations between the grid points. For this purpose, $\tilde{\Sigma}$ is localized by multiplying it point-wise with a smooth correlation function $G(r)$ with exponentially vanishing correlations with distance (Carrassi et al., 2018):

$$\Sigma(r) = \tilde{\Sigma} \circ G(r). \tag{9}$$

Here, $G$ is the numerically efficient Gaspari-Cohn function (Gaspari and Cohn, 1999). This function mimics a Gaussian distribution but vanishes beyond two times the localization radius $L$:

$$G(r) = \begin{cases} 1 - \frac{5}{3} \cdot r^2 + \frac{5}{8} \cdot r^3 + \frac{1}{2} \cdot r^4 - \frac{1}{4} \cdot r^5, & \text{if } 0 \leq r < 1, \\ 4 - 5 \cdot r + \frac{5}{3} \cdot r^2 + \frac{5}{8} \cdot r^3 - \frac{1}{2} \cdot r^4 + \frac{1}{12} \cdot r^5 - \frac{2}{3} \cdot r^{-1}, & \text{if } 1 \leq r < 2, \\ 0, & \text{if } r \geq 2, \end{cases} \tag{10}$$

with $r = \frac{d}{L}$ and $d$ the geographical distance between two grid points.





An AR process is fit at each grid point by means of maximum likelihood. BIC is employed to select the order of the AR
processes with the maximum order set to 4. In a subsequent step, the localization radius to regularize the empirical covariance
matrix of the innovations with is determined by cross-validation. For this purpose, the time series of 230 years is split into
5 folds. Each fold contains 1 block of 6 years and 8 blocks of 5 years spread out evenly across the ESM run with maximum
spacing in time between individual blocks. The empirical covariance matrix of the innovations is estimated based on 4 folds

and regularized with localization radii between 1000 and 4750 km every 250 km. The likelihood to draw the innovations of the
5[th] fold from each one of these regularized covariance matrices is computed. The likelihood values are summed up over all
folds and finally, the localization radius with the maximum likelihood is chosen.

### 4.3   Evaluating the emulator

To evaluate the emulator, the focus is set on ensemble reliability, i.e., the ability to capture the distribution of ESM runs with an

ensemble of emulations (Weigel, 2012). This approach is chosen because the generated emulations cannot be directly compared
to ESM runs as they differ in each realization and thus the traditionally employed measure of "best-fit", i.e., least deviation from
the ESM run, is not a meaningful metric.

Visual as well as quantitative reliability verification are carried out. The visual verification consists of a comparison between
ESM runs and emulations in terms of temperature field snapshots and regionally averaged temperature time series. The time

series are always shown for three specific regions (global land, Central Europe (CEU), and Southern South America (SSA))
which differ substantially in their temperature properties and thus highlight the emulator's flexibility in adapting to regional
climate characteristics. In the quantitative verification, the emulations' ability to reliably reproduce a set of ESM quantiles (5 %,
50 %, 95 %) in the 26 considered SREX regions and global land is evaluated. For this purpose, an ensemble of 200 emulations
is generated for each considered climate model and the percentage of time slots during which the regional temperature in an

ESM run is higher than the respective quantile in the ensemble of emulations is counted. The difference between the emulated
and the counted quantile then gives an indication of the reliability of the emulated ensemble with respect to the ESM run at
hand.

In-sample verification on the training run is conducted for each one of the 40 CMIP5 models. Additionally, out-of-sample
verification on ESM runs not seen during training is carried out for the 12 CMIP5 models with several initial-condition ensem-

ble members. In the latter case, the differences between the emulated and counted quantiles are computed individually for each
ESM initial-condition ensemble member not seen during training and are subsequently averaged.

## 5   Exploring emulator properties for selected ESMs

### 5.1   Calibration results

The calibrated parameters obtained from training an emulator on each one of the four selected ESMs reveal distinct inter-ESM

differences in every emulator module (Fig. 3). The deterministic global mean temperature trends $T_t^{glob,det}$ diverge by about



**Figure 3.** Emulator calibration parameters (rows) for the four selected ESMs (columns). (a) For the global mean temperature emulator module the deterministic global trend $T_t^{glob,det}$ and the AR coefficients plus the standard deviation of the innovations of the global variability $T_t^{glob,var}$ are depicted. (b) For the local mean response module, the regression coefficients are shown. (c) For the local residual temperature variability properties, the AR process order, the standard deviation of the innovations of the AR processes and the localization radius are displayed.





1 °C by the end of the 21st century. For each ESM, the global temperature variability $T_t^{glob,var}$ is described by oscillating AR coefficients with the first lag being positive, but the AR process order p and the standard deviations of the innovations vary between them.

In the local mean response module (Eq. 7), the strongest warming rates, i.e., the largest $\beta_s^{det}$ terms, are found in the northern
high latitudes, but there are substantial differences in the spatial patterns of $\beta_s^{det}$ between emulators trained on different ESMs (Fig. 3). For example, the CESM1(CAM5) emulator warms less in the tropics than the others. The $\beta_s^{var}$ fields indicate that Alaska, Amazon, Africa, and Australia frequently co-vary with global variability $T_t^{glob,var}$. Only in the HadGEM2-ES emulator Central Asia emerges as a region of large $\beta_s^{var}$ values. In each one of the calibrated emulators, the intercept term $\beta_s^{int}$ is generally small in magnitude and smooth in space hinting at overall successful fits.
To model local variability (Eq. 8), AR processes of order 1 or even 0 suffice to capture the characteristic behaviour of each ESM in large parts of the globe (Fig. 3). Generally, there is less memory in the northern high latitudes than in the tropics. The innovations of the local AR processes are largest in magnitude in high latitude continental climates such as North Asia and smallest in the tropics. However, also for these quantities the spatial patterns differ between emulators calibrated on different ESMs. The localization radii chosen to regularize the empirical spatial covariance matrix of the innovations vary between
1500 and 2250 km.

## 5.2   Example realizations

Emulated temperature fields are visually indistinguishable from ESM runs that were not used during training (Fig. 4). All fields exhibit the strongest warming as well as variability in the northern high latitudes. In terms of variability, CESM1(CAM5), HadGEM2-ES, and their emulations, show more patchy behaviour, i.e., locally more confined variability, than CanESM2 and
MPI-ESM-LR.

Time series of emulations and ESM runs not employed during training averaged over global land, CEU, and SSA highlight the emulators capability to reproduce regionally characteristic behaviour of the climate system (Fig. 5). These regions differ in terms of underlying warming trend and variability around this trend. The variability term is smallest on the global scale since local anomalies tend to average out globally. In CEU, the warming rate as well as the variability around the trend are larger
than in SSA.

## 5.3   Emulator transferability between ESMs

Figure 6 shows explicitly what the results of Sects. 5.1 and 5.2 have already hinted at, namely that an the ensemble of emulations generated by an emulator calibrated on a specific ESM is capturing unique properties of that ESM, which in turn are not transferable to other ESMs. For example, the warming rate of the ensemble generated by the CESM1(CAM5) emulator is
inconsistent with ESM runs from all three other ESMs on the global land scale. As expected, differences are also found in the variability around the trend which is, e.g., visibly smaller in SSA in the CESM1(CAM5) emulations than in the runs of the other ESMs. The implications of these results are further discussed in Sect. 7.5.





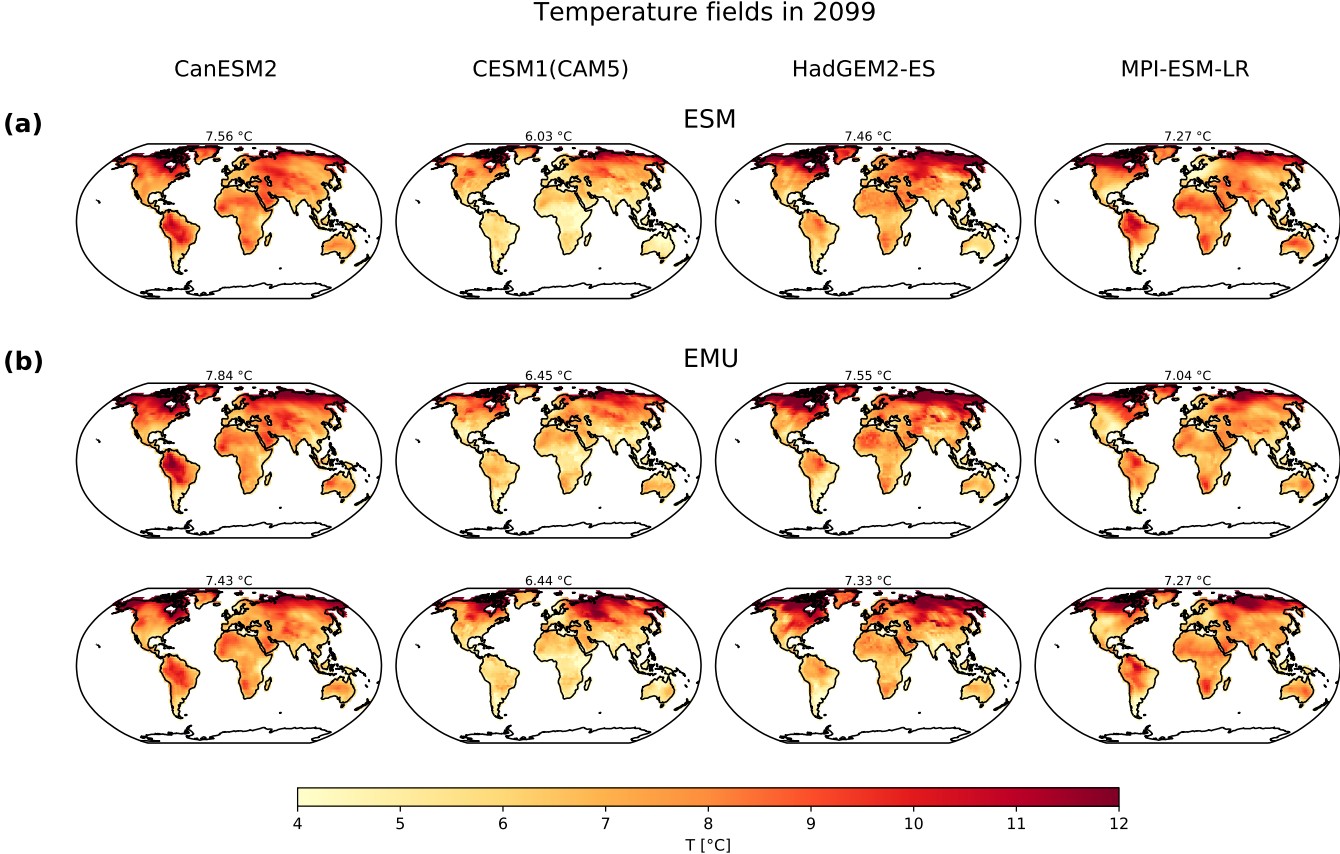

**Figure 4.** Temporal snapshots depicting temperature field realizations in 2099 (rows) for the selected four ESMs (columns). (a) One ESM field not used during training and (b) two corresponding emulations (EMUs) are shown. The temperature on top of each map refers to the global land mean.

## 6   Creating a CMIP5 super-ensemble

Based on the insights gained in Sect. 5, the 40-model CMIP5 ensemble is emulated by training an emulator for each one of the climate models.

### 6.1   Calibration results

Figure 7 shows summary statistics of the calibrated parameters for each CMIP5 climate model highlighting inter-model differences in each emulator module. Additionally, in the supplementary information, plots analogous to Fig. 3 are provided for each climate model for readers interested in the geographical patterns of the calibrated emulator parameters (Figs. S1–S9).

The deterministic global mean temperature trend $T_t^{glob,det}$ at the end of the 21st century ranges between 3.40 and 6.23 °C (Fig. 7). For 45 % of the climate models, the global mean temperature variability $T_t^{glob,var}$ can be modeled as an AR(1) process.

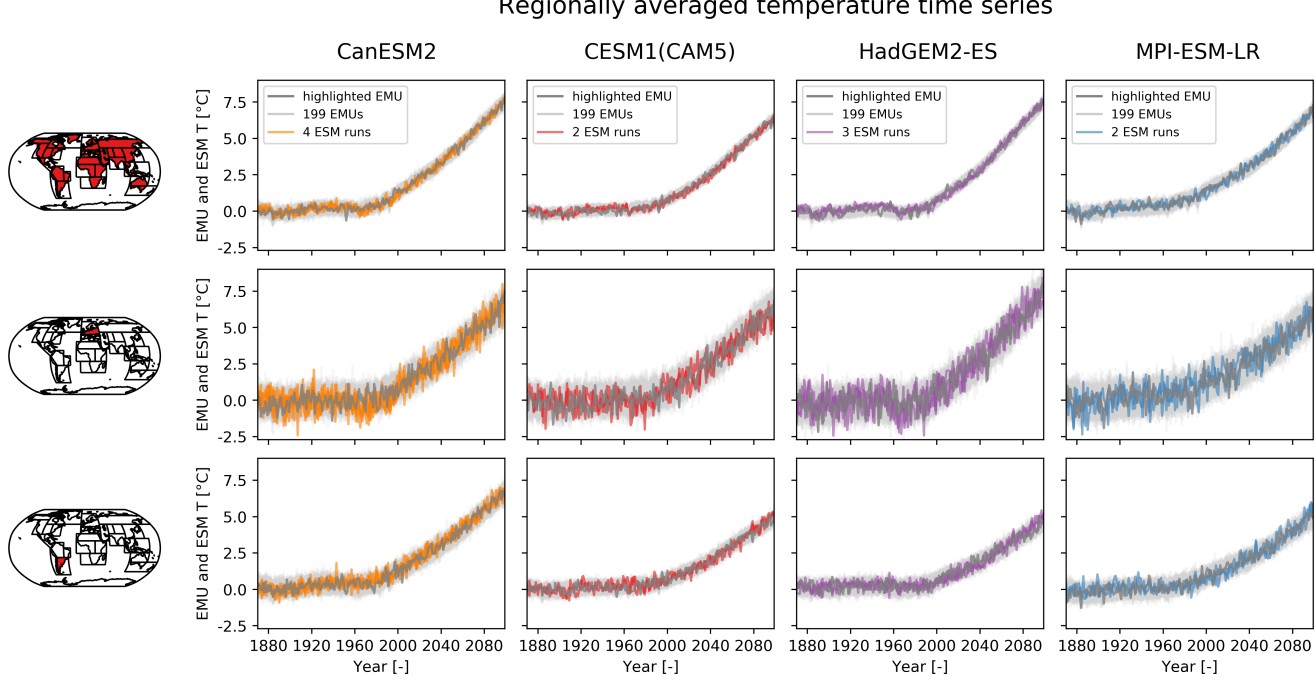

**Figure 5.** Regionally averaged temperature time series (rows) for the four selected ESMs (columns). The regions are from top to bottom: global land, Central Europe (CEU), and Southern South America (SSA). In each panel, 1 emulation (EMU) is highlighted in dark grey and the remaining 199 emulations are shown in light grey. Additionally, all ESM initial-condition ensemble members not employed during training are plotted in color: orange for the four CanESM2 runs, red for the two CESM1(CAM5) runs, purple for the three HadGEM2-ES runs, and blue for the two MPI-ESM-LR runs.

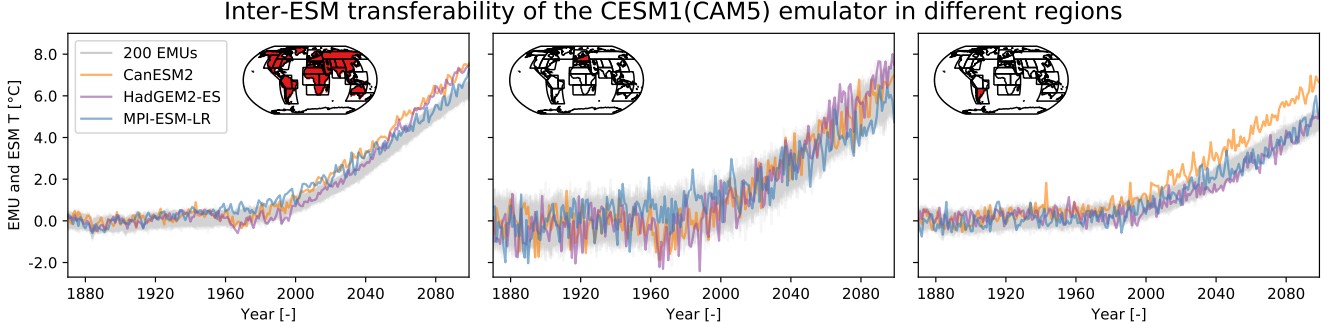

**Figure 6.** Time series of emulations (EMUs) from the CESM1(CAM5) emulator (light grey) overlaid with runs from the three other selected ESMs (CanESM2 orange, HadGEM2-ES purple, MPI-ESM-LR blue) for three regions from left to right: global land, CEU, and SSA.

In the remaining ones either an AR(2) or AR(3) process is chosen. All emulators contain oscillating positive and negative AR

**Figure 7.** Emulator properties (rows) of the 40 CMIP5 climate models (columns). (a) For the global mean temperature emulator, the deterministic signal, the AR coefficients, and the standard deviations of the innovations of the AR process are shown. (b) For the local mean response module, the regression coefficients are depicted. (c) For the local variability emulator, the fraction of grid points for each AR order, the innovations of the AR processes, and the localization radius are depicted. Boxplots indicate the median (dark grey line), the interquartile range (grey box), and the full range of values (grey whiskers).

coefficients with the first coefficient being positive but they differ in the respective magnitude of the AR coefficients. The associated innovations vary in their standard deviations by a factor of almost 3 (0.055–0.148 °C).





In the local mean response module (Eq. 7), more than 50 % of the land grid points warm more quickly than the global mean temperature, i.e., $\beta_s^{det} > 1$, in each calibrated emulator (Fig. 7). In 32 out of 40 calibrated emulators, this is the case in even more than 75 % of the land grid points. Overall, the spread in the $\beta_s^{det}$ terms differs substantially between emulators trained on different climate models. The vast majority of land grid points are positively correlated with global temperature variability $T_t^{glob,var}$, i.e., $\beta_s^{var} > 0$, with the minimum fraction of positive correlations amounting to 82 % of the land grid points. The

intercept terms $\beta_s^{int}$ cluster closely around zero in each calibrated emulator. The fraction of outlier grid points deviating substantially, i.e., $> 1\,°C$, from 0, and hinting at sub-optimal local fits, exceeds 1 % in only 4 of the calibrated emulators and does not exceed 2 % in any.

In the local residual variability (Eq. 8) module, the fraction of grid points with different AR orders varies strongly across the emulators but they agree that considering memory terms of more than one year is only seldom useful (Fig. 7). In fact, for

36 out of 40 models, either an AR(0) or an AR(1) process is chosen in more than 85 % of all grid points. While the median of the standard deviations of the innovations is similar in all calibrated emulators, the full ranges differs, with the maximum lying between $1.36\,°C$ and $2.53\,°C$. The selected localization radii vary between 1500 and 4500 km. Thereby 4500 km is a strong outlier with the second highest selected localization radius amounting to 3000 km. Generally, climate models with a coarser native resolution are associated with larger localization radii (not shown).

## 6.2  Example realizations

Overall, time series of the emulated quantiles follow the original ones closely but are much smoother due to the larger sample size of 8000 realizations as opposed to 40 runs (Fig. 8). However, in some time instances, the actual CMIP5 runs diverge more from the emulated runs indicative of physical processes not accounted for in the emulator set-up. The most notable example occurs approximately between 1960 and 1990 where the median and especially the 5 % quantile of the emulated ensemble are

warmer than the CMIP5 ensemble ones in global land and CEU. This is likely related to a response to tropospheric aerosol forcing which is not accounted in our framework (see discussion in Sect. 7.1). The CMIP5 projections, and thus also the emulations, diverge substantially towards the end of the 21st century in global land and SSA (Fig. 8). In CEU, on the other hand, the modeled warming rates are more similar. As expected, the difference in smoothness between the emulated quantiles and the CMIP5 ones is most pronounced in regions affected by strong local variability such as CEU.

## 6.3  Quantitative verification

### 6.3.1  Approximating CMIP5 – the in-sample verification

To quantitatively assess the emulators' ability to capture runs they were calibrated on, in-sample verification on the training runs is carried out. For the vast majority of CMIP5 models and SREX regions, the median is successfully emulated but the emulations are a bit underdispersive compared to the training run (Fig 9). Generally, the emulations are more reliable for climate

models in which larger localization radii were selected. In North Asia (NAS), said underdispersion is strongest for most models. The only region where the emulations are fully reliable is global land. The underdispersion on the SREX regional scales is





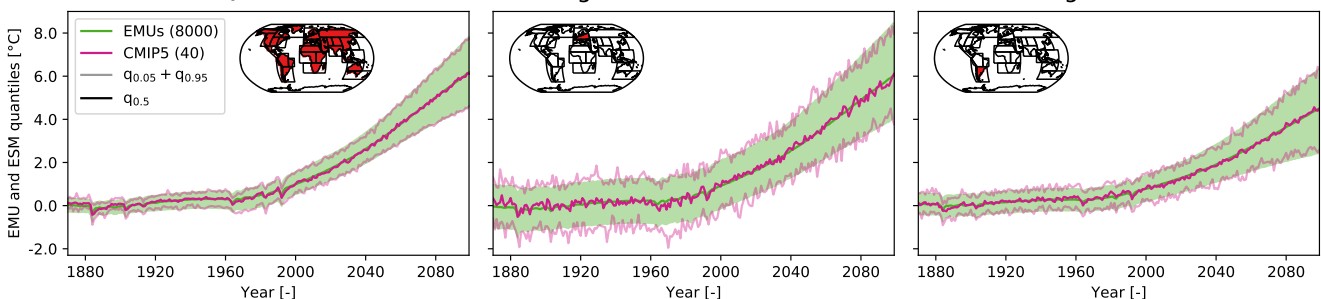

**Figure 8.** Time series of quantiles of the CMIP5 models (one simulation each from 40 models, pink) and their emulations (EMUs, 200 stochastic realizations each from 40 emulators, green) for three regions from left to right: global land, CEU, and SSA. For the EMUs, the median is given in color and the area between the 5th and the 95th quantile is shaded. For CMIP5, the median is shown in color and the lines of the 5th and the 95th quantiles are drawn in light color.

related to regularization and thus inherent to the emulators' design (see discussion in Sect. 7.1). Out of all climate models, the least optimal fit is obtained for MIROC5 with the emulated median being warmer than the one of the training run in many regions.

**6.3.2 Approximating initial-condition ensembles – the out-of-sample verification**

A more challenging task than capturing the training run is mimicking an initial-condition ensemble, the performance of which can be investigated by carrying out an out-of-sample verification with initial-condition ensemble members not seen during training. Qualitatively similar to the the in-sample verification (Sect. 6.3.1), the median is generally well captured but the emulations are a bit underdispersive in regional averages (Fig. 10). However, as expected, the deviations from the emulated

quantiles tend to be larger in magnitude than in the in-sample verification. For most climate models, the strongest deviations in the median are observed in global land, Canada/Greenland/Iceland (CGI), and Southern Australia (SAU). Only for MIROC5, the median of the ensemble of emulations is substantially and systematically warmer than the climate model runs in many regions.

**7 Discussion**

**7.1 Emulator design choices and their effects**

Here, the deterministic global mean temperature trend $T_t^{glob,det}$ (Eq. 4) is approximated by a simple statistical model accounting for smooth forcing with LOWESS smoothing and for abrupt volcanic forcing with a linear regression to stratospheric optical depth, with global variability $T_t^{glob,var}$ rendering the extraction of $T_t^{glob,det}$ challenging. It is assumed that $T_t^{glob,det}$ is shared by all initial-condition ensemble members which is likely not entirely true.





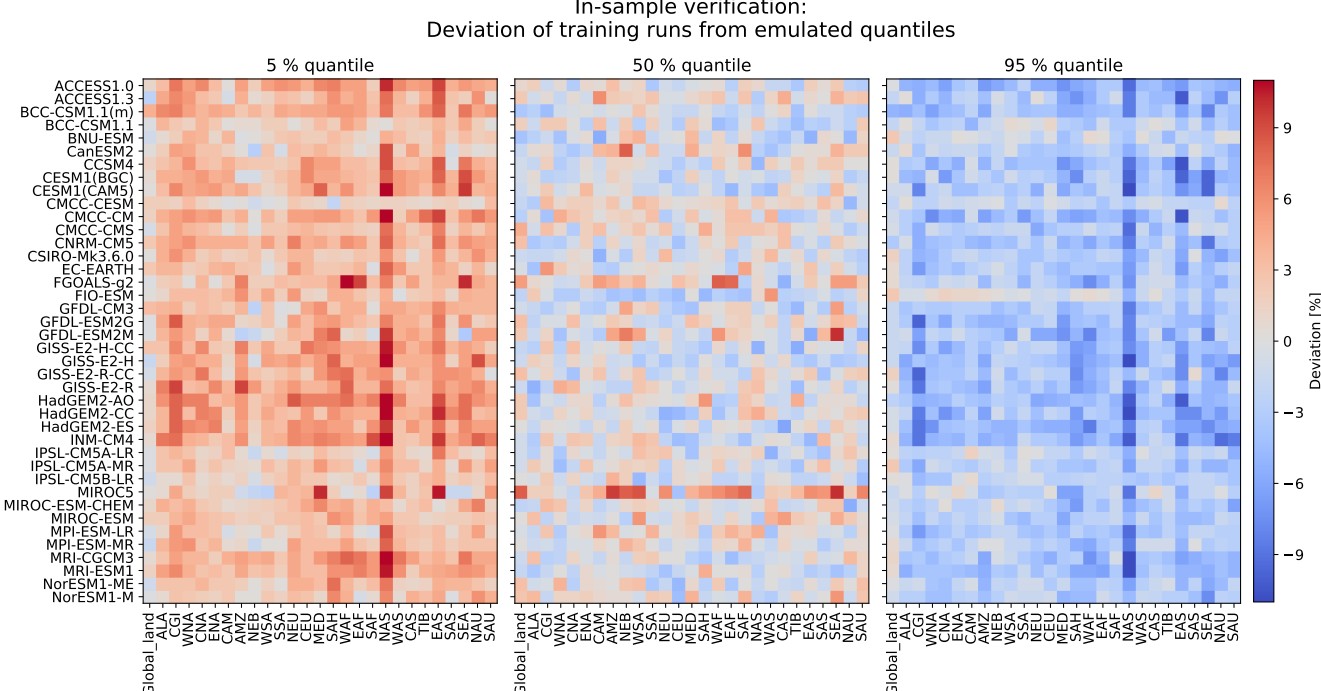

**Figure 9.** 5 % (left), 50 % (middle), and 95 % (right) quantile for the 40 CMIP5 models (rows) and regions (columns). The deviation of the training run from the emulated quantile is given in color. Blue indicates that the emulated quantile is colder than the quantile of the training run, red means that it is warmer.

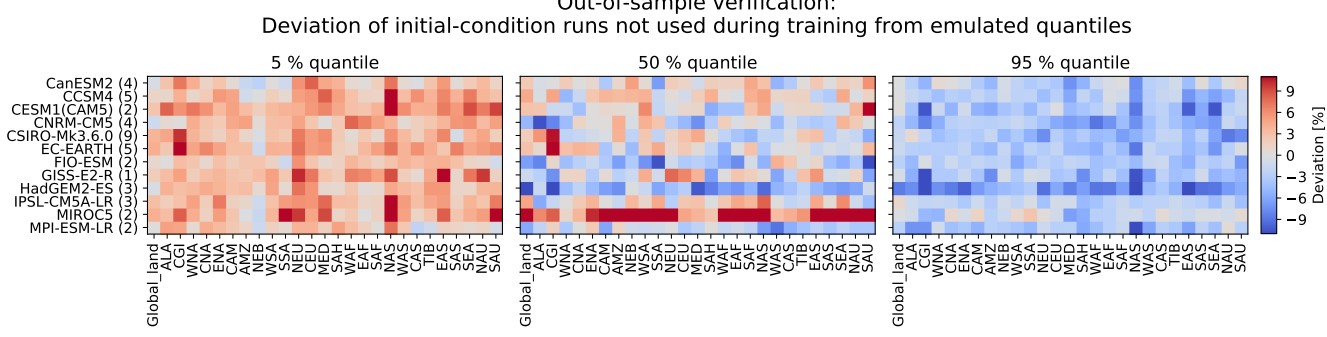

**Figure 10.** 5 % (left), 50 % (middle), and 95 % (right) quantile for the 12 CMIP5 models (rows) providing multiple initial-condition members and regions (columns). The color shows the average deviation across all available initial-condition members which were not used during training. The number of runs averaged over is indicated in brackets behind the model name. Blue indicates that the emulated quantile is colder than the quantile of the climate model runs, red means that it is warmer.





To downscale the global information into a local mean response, a linear approach is chosen (Eq. 7). More complex non-linear methods, specifically, neural networks, were tested as well but resulted in less well-behaved residuals. The overall good performance is in line with the pronounced linear scaling of regional land temperatures with global mean temperature (Seneviratne et al., 2016; Wartenburger et al., 2017) as well as with the widely used linear pattern scaling approaches (Mitchell, 2003; Tebaldi and Arblaster, 2014; Lynch et al., 2017; Osborn et al., 2018). The novelty of our local mean response module

lies in the explicit separation of the global mean temperature response into a response to the deterministic trend $T_t^{glob,det}$ and to the global variability $T_t^{glob,var}$. This separation allows us to better capture the underlying climate phenomena, namely the local warming and the local co-variation with global variability. Because the local mean response is solely conditioned on global temperature, the emulated quantiles are warmer than the CMIP5 model ones between 1960–1990 in certain regions of the world (see Sect. 6.2). During this time, the rapid increase and subsequent decrease in human-induced aerosol emissions

affected surface solar radiation – a phenomenon popularly known as global dimming and brightening – and in turn also surface temperature (Wild, 2012). While our framework could readily be extended to contain tropospheric aerosol predictors, we refrain from it here, as the overall impact on the emulation skill would be small.

In the local residual temperature variability module (Eq. 8), AR processes are fit to account for temporal autocorrelation. If local trends were not successfully removed before fitting the AR processes, they would introduce artificial long-term memory.

However, since AR orders larger than AR(1) are chosen only very rarely, this issue is most likely of minor importance here. A more pronounced effect is caused by regularizing the empirical spatial covariance matrix of the innovations of the local residual temperature variability (Eq. 9) which leads to damped co-variations between grid points and thus underdispersion in regional averages (Sects. 6.3). This explains why the large North Asia region displays the strongest underdisperion and why, generally, more pronounced underdispersion occurs in the CMIP5 models with smaller localization radii, i.e., in the more

strongly regularized ones. Global land is the only region not affected by underdispersion since its variability is driven by the global temperature variability $T_t^{glob,var}$, not by the local innovations $\nu_{s,t}$ which, by construction, average out across the globe. The local residual temperature variability module is based on the assumption that residual local temperature variability is stationary in time which is not fulfilled everywhere in the business-as-usual greenhouse gas emission scenario (see Sect. 2 and Olonscheck and Notz, 2017).

In-sample verification on the training runs and out-of-sample verification on additional initial-condition members not used during training indicate that the assumptions listed above are overall justified since the emulated ensembles generally capture the ESM runs well (Sect. 6.3). Only in 1 climate model, namely MIROC5, the in-sample and especially the out-of-sample verification reveal a systematic warm bias in the median in many regions. Visual inspection of regionally averaged time series (not shown) suggests that this lack-of-fit arises because MIROC5 reacts strongly to human-induced aerosol forcing which we

do not account for.

### 7.2   Climate phenomena emerging from the emulator parameters

On longer time scales, the most pronounced climate phenomena emerging from the emulator's calibration parameters are the stronger warming over land compared to the global mean (Sutton et al., 2007; Hartmann et al., 2013) and the Arctic





amplification (Serreze and Barry, 2011). In the emulator, the enhanced warming rate over land, which is especially large in

the Arctic region, is captured in the regression coefficient $\beta_s^{det}$ of the local mean response model (Eq. 7). Additionally, in line with the Arctic amplification, large variability around the deterministic trend is present in the northern high latitudes in the calibrated emulators.

On shorter time scales, quasi-periodic climate phenomena such as the El Niño Southern Oscillation (Trenberth, 1997) manifest themselves as memory in the global temperature variability $T_t^{glob,var}$ and as characteristic warm and cold anomalies in

regions around the globe. In the emulator, this memory signal is reproduced by modeling $T_t^{glob,var}$ as an AR process (Eq. 6). It is then propagated to the local scale with the local mean response module (Eq. 7). Planetary-scale atmospheric waves, such as Rossby waves (Holton and Hakim, 2013), operate on even shorter time scales and on characteristic, physical-based, length scales. We hypothesize that localization radii below 1500 km are not selected by any of the 40 CMIP5 models because they do not allow to reproduce the large-scale temperature responses induced by planetary-scale atmospheric waves.

### 7.3 The pros and cons of training on single climate model runs

As discussed above, we demonstrate that, for temperature at the spatio-temporal scale considered here, training on a single run per climate model is sufficient to learn key underlying statistical properties of the climate system of this climate model. Early results furthermore indicated that also larger single-model initial-condition ensembles, in that case a 21-member CESM ensemble, can be successfully emulated when training on a single ESM run (Beusch et al., 2018). Since the majority of

modeling groups participating in CMIP5 submitted a single run for the emission pathway considered here, requiring only one run to train the emulator on gives the opportunity to emulate a much larger multi-model ensemble and thus have the resulting super-ensemble account for more inter-model uncertainty. Nevertheless, in order to obtain the best possible emulators to be used e.g., for uncertainty propagation in climate impact or integrated assessment models, it is advisable to employ all available runs for training instead of just a single one for each climate model. Given the assumption that an initial-condition ensemble

shares the statistical properties estimated during the emulator calibration, the more realizations are considered during training, the more robustly underlying trends as well as co-variations between grid points can be estimated.

### 7.4 The advantages of a modular framework

The advantages of the modular framework approach chosen in this study are manifold. First, the calibrated parameters of each emulator module serve as an interesting approach for climate model inter-comparison over a wide range of scales which can

be clearly visualized and easily interpreted (Sects. 5.1 and 6.1). Second, the modular framework renders it straightforward to substitute each emulator module with approaches other than the ones chosen here. For example, with alternative approaches for the deterministic global mean temperature trend (e.g., Meinshausen et al., 2011), for the local mean temperature model (e.g., Tebaldi and Arblaster, 2014; Alexeeff et al., 2018), or for the local residual temperature variability (e.g., Link et al., 2019). Third, if the modeling task were to change, additional predictors could easily be integrated. For example, precipitation

emulation would likely require human-induced aerosol emissions as an additional predictor in the local mean response model (Frieler et al., 2012).



## 7.5 Large single-model initial-condition vs. large multi-model ensembles

Our results highlight fundamental differences between large single-model initial-condition ensembles (Deser et al., 2012; Fischer et al., 2013; Kay et al., 2015; Leduc et al., 2019; Maher et al., 2019) and large multi-model ensembles (Meehl et al., 2007; Taylor et al., 2012; Eyring et al., 2016). While multi-model ensembles are imperfect, with several ESMs showing dependencies (Knutti, 2010; Bishop and Abramowitz, 2013; Sanderson et al., 2015; Abramowitz et al., 2019), multi-model uncertainty nevertheless clearly exceeds single-model initial-condition uncertainty at the yearly scale for temperature (Sect. 5.3). ESMs contained within CMIP5 differ sufficiently across a broad range of scales to sample different phase spaces in projections, which renders it necessary to train an emulator on each climate model to approximate the CMIP5 ensemble. A single-model initial-condition ensemble, on the other hand, can be successfully mimicked by training on a single available ESM run (Sects. 5 and 6.3.2). While beyond the scope of this study, the developed emulator could additionally serve as a novel tool to address the challenge of inter-model dependencies. Differences between climate models could be quantified in terms of their calibrated emulator parameters and subsequently, a subset of models with sufficiently divergent parameters could be selected to base projections on.

## 8 Conclusions and outlook

We introduced a modular framework for climate model emulation of yearly land temperatures and presented a specific, computationally cheap implementation, which makes it possible to create plausible temperature field time series within seconds. Our emulator consists of (i) a global mean temperature module, (ii) a local mean temperature module, and (iii) a local residual temperature variability module. The global mean temperature module contains a deterministic trend as well as a stochastic variability term which is modeled as an AR process. The local mean temperature model is linear in nature and consists of a separate response to the the deterministic global trend and the global variability. The local residual variability module generates spatio-temporally correlated fields by means of locally fit AR processes with spatially correlated innovations.

   With emulators trained on single runs of four selected ESMs, we showed that the emulations are visually indistinguishable from ESM runs not employed during training and that inherent inter-ESM differences in deterministic warming trends and spatio-temporal variability make it necessary to calibrate a separate emulator for each considered climate model. Thus, we proceeded to emulate 40 CMIP5 models by calibrating an emulator on a single run from each climate model. The ensembles of emulations closely resemble the training runs. For CMIP5 models with more than one initial-condition ensemble member, it was furthermore demonstrated that, generally, the ensembles of emulations are able to mimic true climate model initial-condition ensembles. However, a trade-off between parameter estimation robustness and ensemble reliability results in emulations which are, by construction, slightly underdispersive in regional averages. Hence, the generated ensembles overall provide a conservative estimate of regional–scale variability around the mean trend. While the super-ensemble of this study contains 8000 runs – 200 per CMIP5 climate model – it could readily be extended to an arbitrary larger number due to the negligible computational cost of the emulator. Such super-ensembles are expected to be especially helpful in regions with large interannual variability. There, the very noisy quantiles of the CMIP5 ensemble could result in misleading conclusions when

solely employing the CMIP5 ensemble as an input to impact or integrated assessment models which estimate the impact of climate signal uncertainty on their quantity of interest.

Hence, we argue that to sample climate signal uncertainty for yearly temperature at the spatial scale considered in this study, it is more advantageous to invest computational resources in generating multi-model ensembles rather than large single-model ensembles, since the latter can be readily mimicked by our emulator based on a single ESM run. However, when the goal is

to obtain the best possible emulator, we nevertheless advise to train on all available runs with differing initial conditions to enhance the robustness of the estimated statistical parameters. In addition, it could be considered to combine findings from emergent constraints analyses (e.g., Hall and Qu, 2006; Eyring et al., 2019) with the implementation of a climate emulator to derive a super-ensemble based on an observationally-constrained set of ESMs.

The calibrated emulator parameters can be regarded as an ESM-specific "model ID" and provide an interesting avenue for

climate model inter-comparison across a wide range of scales. Inter-model differences can be directly visualized for every emulator module resulting in comprehensible scale-dependent insights into the underlying statistical properties of each climate model.

Future work could focus on introducing additional physical aspects into our statistical emulator by e.g., explicitly modeling modes of interannual variability such as the El Niño Southern Oscillation and the Arctic Oscillation. Furthermore, the emulator

could be extended to simultaneously generate temperature and precipitation fields. Lastly, it would be interesting to investigate how transferable an emulator trained on a specific greenhouse gas emission scenario is to other emission pathways and which modules would need to be modified to account for inter-scenario differences.

*Data availability.* The employed CMIP5 data are available from the public CMIP archive at https://esgf-node.llnl.gov/projects/esgf-llnl/. The stratospheric aerosol optical depth data are provided by NASA and available at https://data.giss.nasa.gov/modelforce/strataer/.

**Appendix A**

*Author contributions.* LB, LG, and SIS designed the study, based on an initial idea from SIS. LB carried out the analysis and drafted the text. LG provided statistical support for the analysis. All authors contributed to interpreting the results and refining the text.

*Competing interests.* The authors declare that they have no conflict of interest.

*Acknowledgements.* We acknowledge partial support from the H2020 CRESCENDO project (grant agreement 641816) and the ERC DROUGHT-
HEAT project (grant agreement 617518). We furthermore acknowledge the World Climate Research Program's Working Group on Coupled





**Table A1.** List of the 40 employed CMIP5 models, the modeling groups providing them, and the number of initial-condition ensemble members used.

| Model | Modeling Center (or Group) | Runs |
|---|---|---|
| ACCESS1.0 | Commonwealth Scientific and Industrial Research Organization (CSIRO) and Bureau of Meteorology (BOM), Australia | 1 |
| ACCESS1.3 | Commonwealth Scientific and Industrial Research Organization (CSIRO) and Bureau of Meteorology (BOM), Australia | 1 |
| BCC-CSM1.1(m) | Beijing Climate Center, China Meteorological Administration | 1 |
| BCC-CSM1.1 | Beijing Climate Center, China Meteorological Administration | 1 |
| BNU-ESM | College of Global Change and Earth System Science, Beijing Normal University | 1 |
| CanESM2 | Canadian Centre for Climate Modeling and Analysis | 5 |
| CCSM4 | National Center for Atmospheric Research | 6 |
| CESM1(BGC) | Community Earth System Model Contributors | 1 |
| CESM1(CAM5) | Community Earth System Model Contributors | 3 |
| CMCC-CESM | Centro Euro-Mediterraneo per I Cambiamenti Climatici | 1 |
| CMCC-CM | Centro Euro-Mediterraneo per I Cambiamenti Climatici | 1 |
| CMCC-CMS | Centro Euro-Mediterraneo per I Cambiamenti Climatici | 1 |
| CNRM-CM5 | Centre National de Recherches Météorologiques / Centre Européen de Recherche et Formation Avancée en Calcul Scientifique | 5 |
| CSIRO-Mk3.6.0 | Commonwealth Scientific and Industrial Research Organization in collaboration with Queensland Climate Change Centre of Excellence | 10 |
| EC-EARTH | EC-EARTH consortium | 6 |
| FGOALS-g2 | LASG, Institute of Atmospheric Physics, Chinese Academy of Sciences and CESS,Tsinghua University | 1 |
| FIO-ESM | The First Institute of Oceanography, SOA, China | 3 |
| GFDL-CM3 | NOAA Geophysical Fluid Dynamics Laboratory | 1 |
| GFDL-ESM2G | NOAA Geophysical Fluid Dynamics Laboratory | 1 |
| GFDL-ESM2M | NOAA Geophysical Fluid Dynamics Laboratory | 1 |
| GISS-E2-H-CC | NASA Goddard Institute for Space Studies | 1 |
| GISS-E2-H | NASA Goddard Institute for Space Studies | 1 |
| GISS-E2-R-CC | NASA Goddard Institute for Space Studies | 1 |
| GISS-E2-R | NASA Goddard Institute for Space Studies | 2 |
| HadGEM2-AO | National Institute of Meteorological Research/Korea Meteorological Administration | 1 |
| HadGEM2-CC | Met Office Hadley Centre | 1 |
| HadGEM2-ES | Met Office Hadley Centre (additional realizations contributed by Instituto Nacional de Pesquisas Espaciais) | 4 |
| INM-CM4 | Institute for Numerical Mathematics | 1 |
| IPSL-CM5A-LR | Institut Pierre-Simon Laplace | 4 |
| IPSL-CM5A-MR | Institut Pierre-Simon Laplace | 1 |
| IPSL-CM5B-LR | Institut Pierre-Simon Laplace | 1 |
| MIROC5 | Atmosphere and Ocean Research Institute (The University of Tokyo), National Institute for Environmental Studies, and Japan Agency for Marine-Earth Science and Technology | 3 |
| MIROC-ESM-CHEM | Japan Agency for Marine-Earth Science and Technology, Atmosphere and Ocean Research Institute (The University of Tokyo), and National Institute for Environmental Studies | 1 |
| MIROC-ESM | Japan Agency for Marine-Earth Science and Technology, Atmosphere and Ocean Research Institute (The University of Tokyo), and National Institute for Environmental Studies | 1 |
| MPI-ESM-LR | Max-Planck-Institut für Meteorologie (Max Planck Institute for Meteorology) | 3 |
| MPI-ESM-MR | Max-Planck-Institut für Meteorologie (Max Planck Institute for Meteorology) | 1 |
| MRI-CGCM3 | Meteorological Research Institute | 1 |
| MRI-ESM1 | Meteorological Research Institute | 1 |
| NorESM1-ME | Norwegian Climate Centre | 1 |
| NorESM1-M | Norwegian Climate Centre | 1 |





Modeling, which is responsible for the Coupled Model Intercomparison Project (CMIP), and we thank the climate modeling groups (listed in Table A1 of this paper) for producing and making available their model output. We are additionally indebted to Urs Beyerle and Jan Sedláček for retrieving and pre-processing the CMIP5 data. Lastly, we would like to thank Julien Brajard, Loris Foresti, and Vincent Humphrey for their valuable input on different modules of our emulator.



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
