# Peer review of "Emulating Earth System Model temperatures with MESMER: from global mean temperature trajectories to grid-point level realizations on land"

_Earth System Dynamics, 2019_

## Short Comment (SC1) · 10 Jul 2019

This paper does some interesting work toward systematizing the way we construct climate model emulators, which could be very useful for comparing different kinds of emulators and for designing interoperable components for emulating climate models.

I would have liked to see a little more depth in section 6.3, "Quantitative verification". The authors show plots comparing the quantiles of the emulator-generated ensemble to the corresponding quantiles of the CMIP ensemble, for three regions, and they remark that "the median [of the CMIP ensemble] is successfully emulated, but the emulations are a bit underdispersive", but this assessment seems to be based entirely

on visual inspection of Figure 8. This analysis would be a lot more compelling if it included quantitative statistical tests, such as a t-test for equality of the means and the Kolmogorov-Smirnov test for equivalence of the overall distributions. If underdispersion is a particular concern, tests for equality of variances could also be applied. Better still would be to develop measures of differences in key properties of the distribution and to derive confidence intervals for those difference measures. Such measures would give prospective users the tools they need to evaluate whether an emulator is fit for whatever use they intend to put it to.

In addition to concerns about how these marginal distributions are evaluated, the marginal distributions appear to be the only dimension along which the authors evaluate the emulator performance. There is no mention at all of testing the spatial correlation or time correlation properties of the emulator. This is a significant omission because the marginal distributions are surely the easiest part to get right when designing an emulator. Capturing the space and time correlations is the true test of the algorithm. In particular, we know that both ESMs and the real climate system display long-range teleconnections and quasi-periodic oscillatory behavior with periods ranging from years to decades. In order to truly evaluate the emulator algorithm, the authors need to explore its ability to produce these phenomena.

The authors' choice to do out of sample validation was interesting, but I am unsure as to whether I agree that it's a useful step in this sort of work. Out of sample validation is normally done when developing models that provide point estimates of the system they are modeling. The theory is that the fitting data is a combination of features that are a deterministic function of the covariates and random features that are idiosyncratic to the sample data. Out of sample validation provides a way to ensure that the model is capturing the former and ignoring the latter.

The goal of this kind of emulator, however, is something different. Instead of trying to provide a point estimate that reflects the influence of certain covariates, we are trying to simulate random draws from the probability distribution implicitly defined by the ESMs,

including all components, both random and deterministic. Therefore, it is not clear what it is that we are trying to exclude by doing out of sample validation. In other words, normally overfitting is caused by the presence of noise (i.e., random response) in the fitting data, but if the noise itself is what we are trying to fit (i.e., we are trying to produce a stochastic variable with similar properties to the noise), then what is it that we are potentially overfitting?

In equation (3) the authors split the global mean temperature time series into a deterministic component and a stochastic variable component. Their purpose in doing this is to allow the local temperature to respond differently to the two components, an innovative approach that makes some sense theoretically. However, they do not take the next step of evaluating the local mean temperature model to see whether the additional coefficient is supported by the data. Either the deviance information criterion (DIC) or Watanabe-Akaike information criterion (WAIC) would be a good choice for such an analysis.

The more I read of the literature in the this area of including variability in climate model emulators, the more I am convinced that designing a plausible emulation algorithm is the easy part of this kind of research. What is hard is proving that the statistical properties of the distribution of the emulator outputs are consistent with those of the emulated system. The big frontier in this research area lies in finding ways to characterize similarities and differences between the joint probability distribution of the variables produced by the emulator and that of the system being emulated. Such methods should be fully quantitatvie (i.e., they should produce a measurement of how much the emulator distribution might deviate from the distribution in the real system). Determining what properties of the joint distribution should be reproduced will be an important step in this sort of evaluation. These properties should include, at a minimum, not only marginal distributions, but also space and time correlation properties.

---

## Referee Comment (RC1) · Anonymous Referee #1 · 6 Aug 2019

General comments

The authors propose a statistical model for emulating output from Earth System Models (ESMs). The model is composed of deterministic and stochastic components that are intended to capture the forced trend and variability, respectively.

There is clearly a lot of work to be done in developing cheap tools like emulators to get more information from our climate model archive, and I am glad to see another contribution to this field. However, I have a number of concerns about the model formulation and, echoing Comment 1 from Robert Link, the validation of the emulator output.

Specific comments

1. One of the challenges of fitting emulators to data or climate model output is separation of the forced and internal components (under the common assumption that they are linearly separable). The authors propose the use of a common approach of regressing onto a smoothed version of the global mean temperature (plus volcanic bursts), but do not provide evidence that this approach is successful. The method can and should be tested within one or multiple initial condition ensembles.

2. The spatial model for the innovations is presented with minimal justification. How was the exponential covariance model chosen versus one that is smoother in space? More importantly, given that the spatial structure of temperature variability depends on the prevailing wind direction, the geometry of the coasts, land surface type, etc., is an isotropic covariance model even appropriate?

3. Identifying parsimonious but sufficient metrics for validation of model ensembles is a challenging and unsolved problem. However, the authors are too qualitative in their evaluation of their emulator skill, which is composed primarily of visual inspection of emulated fields and plots like Figs. 9 and 10. Given the choice of spatial model discussed in (2), it would be helpful to see validation metrics on both the spatial and temporal correlation structure. In addition, the assumption of Gaussianity is built-in but never checked. Finally, validation metrics should be provided with respect to a reasonable null hypothesis, otherwise it is difficult to assess whether a certain error value is meaningful. For example, how large would a given error metric be if different realizations of an actual ESM were resampled, and then the metric of interest was calculated?

4. The writing could be improved to make the manuscript flow more smoothly. In particular, Section 2 could be reworked to more clearly identify what is missing in the current literature that the authors aim to ameliorate with this manuscript.

5. Lines 437-439 make strong statements about replacing single model ensembles with emulators such as the one proposed. Without further validation, I don't think the

authors can say "the latter can be readily mimicked by our emulator based on a single ESM run."

Technical corrections/minor points

1. There are minor grammatical and spelling errors throughout.

2. In the discussion of the forced component, the authors should additionally reference the various LIM-based methods (e.g. Frankignoul et al, 2017, Estimation of the SST Response to Anthropogenic and External Forcing and Its Impact on the Atlantic Multidecadal Oscillation and the Pacific Decadal Oscillation), signal to noise maximizing EOFs (e.g. Ting et al., 2009, Forced and Internal Twentieth-Century SST Trends in the North Atlantic), and low frequency component analysis (e.g. Wills et al., 2018, Disentangling Global Warming, Multidecadal Variability, and El Niño in Pacific Temperatures).

3. The citation of McKinnon and Deser (2018) is slightly misleading. The longer timescales related to coupled modes are explicitly modeled, such that the remaining variability has near-zero memory, and so can be block bootstrapped.

4. I was somewhat mystified by the comment on Line 378 that CMIP5 models do not reproduce the large-scale temperature response to atmospheric waves, which is incorrect. Any reasonable atmospheric model produces Rossby waves and is reasonably accurate at simulating the temperature response.

---

## Referee Comment (RC2) · Anonymous Referee #2 · 20 Aug 2019

General comments

This study developed a modular framework for statistically emulating the CMIP5 ESM global spatially and temporally correlated yearly land temperature field time series. It is shown that this emulator framework can reasonably well emulate the median of the single-ESM initial-condition ensembles of yearly temperature at a negligible computational cost, and with separate emulators calibrated for 40 individual ESMs of the CMIP5, can generate a large ensemble that closely resemble a multi-ESM initial-condition ensemble.

However the visual and quantitative verification of the emulations from this modular

framework have demonstrated that further improvement in the emulators' design is needed before it can be used for the main focus of this study, of using the emulated large ensemble for initializing the climate impact and integrated assessment models used for quantifying the uncertainties in the possible climate system's responses to specific GHG emission pathways. In particular the main concerns are related to the non-dependency of the emulated future projections on the historic ESM performance, the need for more detailed evaluation of the emulation performance, and that the local grid-point scaling reduces the inter-model spread or suppresses the internal variability compared to the ESMs of CMIP5.

Specific comments

1. The computationally cheap statistical emulator developed in this study aims to produce realizations which closely resemble initial-condition ensemble members of the considered ESMs. It will be useful to make a statement on whether it is expected that the future warming projected by each emulation member will be affected by the known differences between observed and simulated historical climate in the ESM considered for calibrating each modular emulator framework. An earlier study cited by the authors (Goodwin, 2016), using large ensemble from an efficient ESM, had suggested that part of the upper range of twenty-first century CMIP5 warming projections may reflect historical simulation–observation inconsistencies.

2. The ability of the ensemble of emulations to capture the distribution of ESM runs is evaluated in this study. It would have been useful to additionally evaluate the performance of these emulations in comparison with the pattern scaling approach used to relate the grid-scale local temperature linearly with the global mean temperature. An earlier study cited by the authors (Castruccio et al., 2014), demonstrated using a lack-of-fit statistic that the emulated local annual temperatures outperformed pattern scaling at grid point scales, and captures the nonlinear evolution of spatial patterns of climate anomalies inherent in transient climates. A similar evaluation could have been used in the present study to compare the measured emulation performance with a simple

global pattern scaling.

3. The quantitative verification in section 6.2 brought out that although the CMIP5 ESM median is emulated well, the emulations are underdispersive compared to the training run, particularly at regional scales. This issue is identified to be related to the emulators' design. Although the authors conclude that this underestimation is the result of a trade-off between parameter estimation robustness and ensemble reliability, it will be challenging to expect that the climate impact and integrated assessment models initialised with these emulations will be able to infer properties of the existing CMIP5 multi-ESM initial-condition ensemble runs which have not been generated yet.

Technical comments

1. Section 2 introducing the proposed modular emulation framework in the context of previous studies may be merged into Section 4 describing this additive framework with three sub-modules so that the extent to which the previously developed methods were utilised in the implementation of this modular framework can be brought out clearly.

2. Line 353, please correct "underdispersion".

---

## Author Comment (AC1) · 19 Sep 2019

**Answer to Robert Link**

We thank Robert Link for taking the time to write such a detailed short comment on our paper and for raising several points which will help to improve the quality of the manuscript. In the following, we provide a point-by-point answer whereby we show R. Link's comment in black and our response in blue.

This paper does some interesting work toward systematizing the way we construct climate model emulators, which could be very useful for comparing different kinds of emulators and for designing interoperable components for emulating climate models.

We are happy to hear that our effort to systematize the design of emulators and provide a modular framework is appreciated.

I would have liked to see a little more depth in section 6.3, "Quantitative verification". The authors show plots comparing the quantiles of the emulator-generated ensemble to the corresponding quantiles of the CMIP ensemble, for three regions, and they remark that "the median [of the CMIP ensemble] is successfully emulated, but the emulations are a bit underdispersive", but this assessment seems to be based entirely on visual inspection of Figure 8. This analysis would be a lot more compelling if it included quantitative statistical tests, such as a t-test for equality of the means and the Kolmogorov-Smirnov test for equivalence of the overall distributions. If underdispersion is a particular concern, tests for equality of variances could also be applied. Better still would be to develop measures of differences in key properties of the distribution and to derive confidence intervals for those difference measures. Such measures would give prospective users the tools they need to evaluate whether an emulator is fit for whatever use they intend to put it to.

A misunderstanding seems to have occurred here. Figure 8 is still part of Sect. 6.2 "Example realizations". Figures 9 and 10, however, provide a quantitative verification for all climate models in all SREX regions for the 5th, the 50th, and the 95th quantile with Fig. 9 depicting the performance on the training runs and Fig. 10 the performance on independent initial-condition ensemble members not seen during training. These analyses address the main concerns of the reviewer and show that the developed emulator has a satisfactory performance in the representation of an initial-condition ensemble despite being trained on a single run.
Furthermore, we agree on the usefulness of additional quantitative verification and will adapt the revised manuscript accordingly. More specifically, Figs. 9 and 10 will be extended to contain information about the degree to which true ESM runs are indistinguishable from single emulated runs.

In addition to concerns about how these marginal distributions are evaluated, the marginal distributions appear to be the only dimension along which the authors evaluate the emulator performance. There is no mention at all of testing the spatial correlation or time correlation properties of the emulator. This is a significant omission because the marginal distributions are surely the easiest part to get right when designing an emulator. Capturing the space and time correlations is the true test of the algorithm. In particular, we know that both ESMs and the real climate system display long-range teleconnections and quasi-periodic oscillatory behavior with periods ranging from years to decades. In order to truly evaluate the emulator algorithm, the authors need to explore its ability to produce these phenomena.

As noted by R. Link, there was no explicit evaluation of the space-time characteristics of the emulator in the original manuscript. In the revised manuscript, we will include additional analyses showing to what extent serial and spatial correlations of ESM runs can be retained by our simple statistical emulator.

However, it should be pointed out that Figs. 9 and 10 already constitute a very aggregated form of a space-time verification. These figures show the skill of the emulator in capturing the underlying trend (as indicated by $50^{th}$ percentile) as well as the variability around it (as indicated by the $5^{th}$ and the $95^{th}$ percentile) on regional scales. Large parts of the regional variability are reproduced but the emulations are slightly underdispersive compared to true ESM runs. Without accounting for spatial correlation in the innovations of the local residual variability module, the results would be far more underdispersive. Furthermore, we would like to note that the scope of the present study was to focus on the generating stochastic fields that resemble temperature at regional scales based on a single training run, not on emulating far-reaching teleconnections and multi-decadal variability. The considered local residual variability module was chosen accordingly and is by design not able to reproduce teleconnections at the planetary scale or multi-decadal dependencies. In the revised manuscript, we will clarify these design choices alongside limitations with respect to spatial teleconnections and multi-decadal dependencies.

The authors' choice to do out of sample validation was interesting, but I am unsure as to whether I agree that it's a useful step in this sort of work. Out of sample validation is normally done when developing models that provide point estimates of the system they are modeling. The theory is that the fitting data is a combination of features that area deterministic function of the covariates and random features that are idiosyncratic to the sample data. Out of sample validation provides a way to ensure that the model is capturing the former and ignoring the latter.

The goal of this kind of emulator, however, is something different. Instead of trying to provide a point estimate that reflects the influence of certain covariates, we are trying to simulate random draws from the probability distribution implicitly defined by the ESMs, including all components, both random and deterministic. Therefore, it is not clear what it is that we are trying to exclude by doing out of sample validation. In other words, normally overfitting is caused by the presence of noise (i.e., random response) in the fitting data, but if the noise itself is what we are trying to fit (i.e., we are trying to produce a stochastic variable with similar properties to the noise), then what is it that we are potentially overfitting?

Based on the comments of R. Link and reviewer 1, we realized that the chosen "out-of-sample" terminology may have been misleading for some readers. To increase clarity, we will no longer employ the "out-of-sample" term in the revised manuscript, instead we will explicitly refer to "independent initial-condition ensemble members not employed during training".

Note also that we regard testing the emulator's performance on independent initial-condition ensemble members, where possible, essential because training on a single run might results in overfitting to this specific realization. Our analyses reveal a satisfactory performance of the emulator on regional scales when evaluated against both the training run and independent initial-condition ensemble members not seen during training. Please refer to Sect. 6.3.2. of the original manuscript for a detailed discussion on this topic.

In equation (3) the authors split the global mean temperature time series into a deterministic component and a stochastic variable component. Their purpose in doing this is to allow the local temperature to respond differently to the two components, an innovative approach that makes some sense theoretically. However, they do not take the next step of evaluating the local mean temperature model to see whether the additional coefficient is supported by the data. Either the deviance information criterion (DIC) or Watanabe-Akaike information criterion (WAIC) would be a good choice for such an analysis.

In the paper, we have decided to follow the approach of proposing a specific implementation for our emulator without testing individual components against alternatives as there would be countless alternative implementations which could be considered. In the revised manuscript, we will emphasize that the presented modular framework allows to exchange individual components to accommodate specific user needs.
Note also that we have tested both regression against the full global mean temperature time series and regression against the split-up global mean temperature time series in the exploratory phase of this study. The chosen configuration led to less underdispersive results on regional scales.

The more I read of the literature in the this area of including variability in climate model emulators, the more I am convinced that designing a plausible emulation algorithm is the easy part of this kind of research. What is hard is proving that the statistical properties of the distribution of the emulator outputs are consistent with those of the emulated system. The big frontier in this research area lies in finding ways to characterize similarities and differences between the joint probability distribution of the variables produced by the emulator and that of the system being emulated. Such methods should be fully quantitatvie (i.e., they should produce a measurement of how much the emulator distribution might deviate from the distribution in the real system). Determining what properties of the joint distribution should be reproduced will be an important step in this sort of evaluation. These properties should include, at a minimum, not only marginal distributions, but also space and time correlation properties.

We appreciate this philosophical input on emulator research. We agree that it is challenging to find the most suitable validation metrics for emulators which contain variability. Especially, because it heavily depends on the application in mind which validation metrics are even suitable to look at. For example, for researchers interested in impacts on regional scales, such as the ones caused by heat waves, it may be more important to reproduce local correlations than far-reaching teleconnections. Additionally, even though fully quantitative validation metrics can be very informative, one should not underestimate the importance of intuitive validation strategies, which can be easily understood by potential users that may have a less technical background. For example, demonstrating an emulator's capability to produce visually consistent output with ESMs can be very helpful in communicating an emulator's worth to a broader audience of potential users.

---

## Author Comment (AC2) · 19 Sep 2019

**Answer to Anonymous Referee #1**

We thank the anonymous referee for the constructive feedback which will help to improve the quality of our manuscript. In the following, we provide a point-by-point answer to the reviewer whereby we show the reviewer's comment in black and our response in blue.

**General comments**

The authors propose a statistical model for emulating output from Earth System Models (ESMs). The model is composed of deterministic and stochastic components that are intended to capture the forced trend and variability, respectively. There is clearly a lot of work to be done in developing cheap tools like emulators to get more information from our climate model archive, and I am glad to see another contribution to this field. However, I have a number of concerns about the model formulation and, echoing Comment 1 from Robert Link, the validation of the emulator output.

We are happy to hear that the reviewer agrees that developing computationally cheap tools to get more information from the climate model archive is important. In the following, we will address the concerns the reviewer expresses.

**Specific comments**

1. One of the challenges of fitting emulators to data or climate model output is separation of the forced and internal components (under the common assumption that they are linearly separable). The authors propose the use of a common approach of regressing onto a smoothed version of the global mean temperature (plus volcanic bursts), but do not provide evidence that this approach is successful. The method can and should be tested within one or multiple initial condition ensembles.

We thank the reviewer for this comment. There seems to have been a misunderstanding caused by a naming convention we chose, which resulted in comments from both this reviewer and R. Link. In fact, we do test the emulator using multiple initial-condition ensembles. While we calibrate the emulator with a single run per climate model, for all models where several initial-condition members are available, we evaluate the performance of the emulator using initial-condition members not employed during training in Sect. 6.3.2 of the original manuscript. For illustrative examples, please additionally consult Fig. 5 of the original manuscript. Throughout the manuscript, we referred to this type of evaluation as "out-of-sample" testing.

To improve the readability of the paper, we will exchange the "out-of-sample" terminology with explicitly referring to "independent initial-condition ensemble members not employed during training".

2. The spatial model for the innovations is presented with minimal justification. How was the exponential covariance model chosen versus one that is smoother in space? More importantly, given that the spatial structure of temperature variability depends on the prevailing wind direction, the geometry of the coasts, land surface type, etc., is an isotropic covariance model even appropriate?

A misunderstanding occurred here. We do not employ an exponential covariance model as a spatial model for the innovations, instead we sample from a regularized empirical covariance matrix which is detailed in Eq. 9 of the original manuscript. For the regularization, we employ an approach referred to as localization which is well established in the field of data assimilation (Carrassi et al., 2018). To convey this point more clearly, we will dedicate more text to the justification of our spatial model in the revised manuscript, highlighting that we employ an approach which is common in data assimilation and which is able to retain anisotropy in the underlying data on regional scales.

3. Identifying parsimonious but sufficient metrics for validation of model ensembles is a challenging and unsolved problem. However, the authors are too qualitative in their evaluation of their emulator skill, which is composed primarily of visual inspection of emulated fields and plots like Figs. 9 and 10. Given the choice of spatial model discussed in (2), it would be helpful to see validation metrics on both the spatial and temporal correlation structure. In addition, the assumption of Gaussianity is built-in but never checked. Finally, validation metrics should be provided with respect to a reasonable null hypothesis, otherwise it is difficult to assess whether a certain error value is meaningful. For example, how large would a given error metric be if different realizations of an actual ESM were resampled, and then the metric of interest was calculated?

To address the reviewers call for a more quantitative validation of the emulator, we plan to extend the space-time verification of our emulator in the revised manuscript. Specifically, to address the concerns raised by this reviewer: we will (1) expand the verification section in the paper to include both verification of the deterministic trend and the variability around it, (2) include results from a Shapiro-Wilks test to demonstrate the validity of the Gaussianity assumption of the innovations of the local residual variability in the supplementary material, (3) extend Figs. 9 and 10 of the original paper to contain information on the degree to which true ESM runs are indistinguishable from single emulated runs.

4. The writing could be improved to make the manuscript flow more smoothly. In particular, Section 2 could be reworked to more clearly identify what is missing in the current literature that the authors aim to ameliorate with this manuscript.

The text (in particular Sect. 2) will be carefully revised and re-structured as needed to improve the readability of the manuscript. In particular, we will focus on highlighting the added value of our study compared to existing literature more explicitly.

5. Lines 437-439 make strong statements about replacing single model ensembles with emulators such as the one proposed. Without further validation, I don't think the authors can say "the latter can be readily mimicked by our emulator based on a single ESM run."

As highlighted in our answer to the specific comment 1, there seems to have been a misunderstanding regarding our validation of the ability of the emulator in reproducing initial-condition ensembles. The diagnostics we have provided in Sect. 6.3.2 of the original manuscript address these concerns. But we will improve the clarity of the manuscript with respect to this point and also provide additional validation metrics. This specific sentence will be replaced with a more in-depth assessment of the potential of an emulator such as the one we have developed to achieve this goal.

Technical corrections/minor points

1. There are minor grammatical and spelling errors throughout.

The manuscript will be carefully revised with focus on grammatical and spelling errors.

2. In the discussion of the forced component, the authors should additionally reference the various LIM-based methods (e.g. Frankignoul et al, 2017, Estimation of the SST Response to Anthropogenic and External Forcing and Its Impact on the Atlantic Multidecadal Oscillation and the Pacific Decadal Oscillation), signal to noise maximizing EOFs (e.g. Ting et al., 2009, Forced and Internal Twentieth-

Century SST Trends in the North Atlantic), and low frequency component analysis (e.g. Wills et al., 2018, Disentangling Global Warming, Multidecadal Variability, and El Niño in Pacific Temperatures).

We thank the reviewer for directing us towards LIM-based methods, and we will consider including these discussion points in the revised manuscript.

3. The citation of McKinnon and Deser (2018) is slightly misleading. The longer timescales related to coupled modes are explicitly modeled, such that the remaining variability has near-zero memory, and so can be block bootstrapped.

We thank the reviewer for noting this, and we will revise the text accordingly.

4. I was somewhat mystified by the comment on Line 378 that CMIP5 models do not reproduce the large-scale temperature response to atmospheric waves, which is incorrect. Any reasonable atmospheric model produces Rossby waves and is reasonably accurate at simulating the temperature response.

We thank the reviewer for pointing out that the corresponding line was not formulated clearly enough. In the revised manuscript, we will clarify that: "We hypothesize that localization radii below 1500 km are not selected in any of the 40 CMIP5 model emulators, because such localization radii create too small-scale stochastic temperature variability which cannot mimic typical temperature responses induced by planetary-scale atmospheric waves in climate models."

---

## Author Comment (AC3) · 19 Sep 2019

**Answer to Anonymous Referee #2**

We thank the anonymous referee for the helpful suggestions which will lead to an improved manuscript. In the following, we provide a point-by-point answer to the reviewer whereby we show the reviewer's comment in black and our response in blue.

General comments
This study developed a modular framework for statistically emulating the CMIP5 ESM global spatially and temporally correlated yearly land temperature field time series. It is shown that this emulator framework can reasonably well emulate the median of the single-ESM initial-condition ensembles of yearly temperature at a negligible computational cost, and with separate emulators calibrated for 40 individual ESMs of the CMIP5, can generate a large ensemble that closely resemble a multi-ESM initial-condition ensemble.
However the visual and quantitative verification of the emulations from this modular framework have demonstrated that further improvement in the emulators' design is needed before it can be used for the main focus of this study, of using the emulated large ensemble for initializing the climate impact and integrated assessment models used for quantifying the uncertainties in the possible climate system's responses to specific GHG emission pathways. In particular the main concerns are related to the non-dependency of the emulated future projections on the historic ESM performance, the need for more detailed evaluation of the emulation performance, and that the local grid-point scaling reduces the inter-model spread or suppresses the internal variability compared to the ESMs of CMIP5.

Each one of the concerns raised by the reviewer in this introduction is explained in more detail by the reviewer in the specific comments below. Hence, we directly answer these points below.

Specific comments
1. The computationally cheap statistical emulator developed in this study aims to produce realizations which closely resemble initial-condition ensemble members of the considered ESMs. It will be useful to make a statement on whether it is expected that the future warming projected by each emulation member will be affected by the known differences between observed and simulated historical climate in the ESM considered for calibrating each modular emulator framework. An earlier study cited by the authors (Goodwin, 2016), using large ensemble from an efficient ESM, had suggested that part of the upper range of twenty-first century CMIP5 warming projections may reflect historical simulation–observation inconsistencies.

In this study, we provide a computationally cheap machinery which can take a run from any ESM it is provided with and generate additional realizations closely resembling an initial-condition ensemble. Thus, it is explicitly not part of the tasks of our emulator to judge the ESM simulations. Instead, the emulator is designed to be flexible enough to emulate whatever ESM run it is provided with. The idea behind this is that users may identify a subset of ESMs most suitable for their task at hand and then have the opportunity to generate additional realizations which approximate an initial-condition ensemble of this subset of ESMs. Consequently, validation of ESMs does not lie within the scope of the present paper. To clarify this, we will include statements stressing that we only focus on emulation here, not on model performance and its implications for the realism of the projections.

2. The ability of the ensemble of emulations to capture the distribution of ESM runs is evaluated in this study. It would have been useful to additionally evaluate the performance of these emulations in comparison with the pattern scaling approach used to relate the grid-scale local temperature linearly

with the global mean temperature. An earlier study cited by the authors (Castruccio et al., 2014), demonstrated using a lack-of-fit statistic that the emulated local annual temperatures outperformed pattern scaling at grid point scales, and captures the nonlinear evolution of spatial patterns of climate anomalies inherent in transient climates. A similar evaluation could have been used in the present study to compare the measured emulation performance with a simple global pattern scaling.

We thank the reviewer for this suggestion. We will consider including a supplementary comparison of true ESM time series to time series obtained by simple pattern scaling and by our emulator. This will highlight the most important distinction between pattern scaling and our emulation approach, namely that traditional pattern scaling focuses solely on the mean response and does not contain a local variability module. Hence, additional realizations obtained from pattern scaling are by definition far more underdispersive than output from the emulator introduced in this study.

3. The quantitative verification in section 6.2 brought out that although the CMIP5 ESM median is emulated well, the emulations are underdispersive compared to the training run, particularly at regional scales. This issue is identified to be related to the emulators' design. Although the authors conclude that this underestimation is the result of a trade-off between parameter estimation robustness and ensemble reliability, it will be challenging to expect that the climate impact and integrated assessment models initialised with these emulations will be able to infer properties of the existing CMIP5 multi-ESM initial-condition ensemble runs which have not been generated yet.

We would like to thank the reviewer for noting that the added value of the emulated ensemble members has not been shown in sufficient detail. To accommodate for this (and in connection with the call for more space-time validation by other referees), we will include further assessments of the space-time characteristics in the revised manuscript. Among other things, this will show that the emulations reliably reproduce the variability of ESM simulations at the grid-cell level.

Technical comments
1. Section 2 introducing the proposed modular emulation framework in the context of previous studies may be merged into Section 4 describing this additive framework with three sub-modules so that the extent to which the previously developed methods were utilised in the implementation of this modular framework can be brought out clearly.

Please note that we deliberately kept the framework introduction in the context of previous studies (Sect. 2) separate from the detailed study-specific technical implementation (Sect. 4) because we believe that this makes it easier to follow the manuscript. In the revised manuscript, Sect. 7.1 will be expanded to discuss similarities between existing literature and our emulator implementation in more detail.

2. Line 353, please correct "underdispersion".

This is noted and will be done.

---

## Author Response (AR1)

Lea Beusch
Institute for Atmospheric and
Climate Science ETH Zurich
Universitaetstrasse 16
CH-8092 Zurich
Phone: +41  44 633 36 24
E-mail: lea.beusch@env.ethz.ch

[Figure]

Eidgenössische Technische Hochschule Zürich
Swiss Federal Institute of Technology Zurich

Earth System Dynamics Editorial Board

Zurich, 08 November 2019

**Emulating Earth System Model temperatures: from global mean temperature trajectories to grid-point level realizations on land**

Dear Dr. Krishnan,

Please find enclosed the revised version of our manuscript and additionally a version with tracked changes directly attached to this letter. For the answer to the reviewers we refer to the point-by-point answers we posted in the interactive discussion.

Based on the reviewers suggestions and your comments, the following main changes were made to the manuscript:

1. A new terminology has been introduced in which climate model runs used to train the emulator are referred to as training runs and the additional initial-condition ensemble members are referred to as test runs. The new terminology improves the clarity of the manuscript and makes it more apparent that we evaluate our emulations both with respect to the training run and independent initial-condition ensemble members which were not employed during training.

2. The quantitative verification section of the manuscript has been extended substantially. In addition to the regional-scale emulation verification, a separate verification of local trends and local variability is now included with a special focus set on the space-time characteristics of the local variability and on how distinguishable climate model runs are from emulations.

3. A supplementary comparison between emulations and simple pattern scaling results has been added to demonstrate their differences.

4. The added value of our study compared to existing literature is now explicitly stated and the extent to which previously developed methods are used in this study is discussed more extensively.

Please also note that during the review process we have improved the local residual temperature variability module in our emulator. The new module locally fits AR(1) processes instead of AR(p) processes which makes more time slots available to estimate the spatial cross-correlations between grid points and therefore decreases the amount of regularization needed.

For all additional changes we implemented based on the reviews, please refer directly to the revised manuscript.

We are confident that the revisions increased the quality of the manuscript.

Yours sincerely,

Lea Beusch
(on behalf of all co-authors)

**Emulating Earth System Model temperatures: from global mean temperature trajectories to grid-point level realizations on land**

Lea Beusch[1], Lukas Gudmundsson[1], and Sonia I. Seneviratne[1]

[1]Institute for Atmospheric and Climate Science, ETH Zurich, Zurich, Switzerland

**Correspondence:** Lea Beusch (lea.beusch@env.ethz.ch)

**Abstract.** Earth System Models (ESMs) are invaluable tools to study the climate system's response to specific greenhouse gas emission pathways. Large single-model initial-condition and multi-model ensembles are used to investigate the range of possible responses and serve as input to climate impact and integrated assessment models. Thereby, climate signal uncertainty is propagated along the uncertainty chain and its effect on interactions between humans and the Earth system can be quantified. However, generating both single-model initial-condition and multi-model ensembles is computationally expensive. In this study, we assess the feasibility of geographically-explicit climate model emulation, i.e., of statistically producing large ensembles of global [..[1] ]land temperature field time series that closely resemble ESM runs at a negligible computational cost[..[2] ]. For this purpose, we develop a modular [..[3] ]emulation framework which consists of (i) a global mean temperature [..[4] ]module, (ii) a local [..[5] ]temperature response module, and (iii) a local residual temperature variability [..[6] ]module. We first show that to successfully mimic single-model initial-condition ensembles of yearly temperature from 1870 to 2100 on grid-point to regional scales, it is sufficient to train on a single ESM run, but separate emulators need to be calibrated for individual ESMs given fundamental inter-model differences. We then emulate 40 climate models of the Coupled Model Intercomparison Project Phase 5 (CMIP5) to create a "super-ensemble", i.e., a large ensemble [..[7] ]which closely resembles a multi-model initial-condition ensemble. [..[8] ]The thereby emerging ESM-specific emulator [..[9] ]parameters provide essential insights on inter-model [..[10] ]differences across a broad range of scales [..[11] ]and characterize core properties of each ESM. Our results highlight that, for temperature at the spatio-temporal scales considered here, it is likely more advantageous to invest computational resources into generating multi-model ensembles rather than large single-model initial-condition en-
* * *
[1]removed: spatially and temporally correlated

[2]removed: which closely resemble ESM runs spanning from 1870 to 2099.

[3]removed: framework that

[4]removed: emulator

[5]removed: mean temperature emulator

[6]removed: emulator

[7]removed: that

[8]removed: Furthermore, the

[9]removed: calibration

[10]removed: divergences

[11]removed: which can be viewed as a "model ID" of

sembles. Such multi-model ensembles can [..[12] ]be extended to super-ensembles with [..[13] ]emulators like the one presented here.

20  *Copyright statement.*  TEXT

**1  Introduction**

[revised manuscript text omitted]

$$T_{s,t} = f(T_t^{glob}) + \eta_{s,t}, \tag{1}$$

where the [..[31] ]local temperature $T_{s,t}$ at grid point $s$ and time $t$ is defined as a [..[32] ]response to the [..[33] ]global mean temperature [..[34] ]$T_t^{glob}$, indicated by the function $f()$, and a stochastic local residual temperature variability term $\eta_{s,t}$. [..[35] ]Contributions from physical feedbacks other than the ones captured within the global mean temperature signal are thus neglected. The assumption of an underlying additivity is in line with frequently employed approaches in [..[36] ]uncertainty analysis in climate science (Hawkins and Sutton, 2009) and in climate change detection and attribution studies (Allen and Stott, 2003).

Our framework requires three [..[37] ]modules: a global mean temperature [..[38] ]module, a module for the grid-point level temperature response to the global mean temperature, and a local residual temperature variability [..[39] ]module. In the fol-
* * *
[24]removed: the

[25]removed: emulation

[26]removed: proposed

[27]removed: calibration results and example realizations

[28]removed: selected

[29]removed:  5.In Sect. 6, the emulator is applied to a large

[30]removed: In addition to calibration results and example realizations, quantitative verification of in-sample and out-of-sample performance are presented.

[31]removed: temperature $T$

[32]removed: deterministic

[33]removed: current

[34]removed: $T_t^{glob}$

[35]removed: Deterministic contributions

[36]removed: climate science uncertainty analysis (Hawkins and Sutton, 2009) and

[37]removed: sub-modules. An emulator for

[38]removed: , an emulator for the deterministic mean

[39]removed: emulator

[revised manuscript text omitted]

[100]removed: Framework implementation

[101]removed: Global mean temperature emulator

[102]removed: The

[103]removed: emulator stochastically creates

[104]removed: . Global mean temperature

[105]removed: deterministic part $T_t^{glob,det}$

[106]removed: stochastic global variability part

[107]removed: varying between realizations:

[109]removed: the deterministic trend, $T_t^{glob,det}$,

185 In a next step, [..[111] ]$T_t^{glob,volc}$ is approximated as the linear response of the residuals of the smooth trend, i.e., [..[112] ]$T_t^{glob} - T_t^{glob,sm}$, to stratospheric aerosol optical depth $AOD_t$ with regression coefficients $\lambda_0$ and $\lambda_1$:

$$T_t^{glob,volc} = \lambda_0 + \lambda_1 \cdot AOD_t \tag{4}$$

The time series of [..[113] ]global mean temperature variability $T_t^{glob,var} = T_t^{glob} - T_t^{glob,trend}$ is modeled as an AR process of order p with coefficients $\alpha_0,...,\alpha_p$ such that

$$\quad T_t^{glob,var} = \alpha_0 + \sum_{k=1}^{k=p} \alpha_k \cdot T_{t-k}^{glob,var} + \epsilon_t \quad \text{with} \quad \epsilon_t \sim \mathcal{N}(0,\sigma), \tag{5}$$

whereby $\epsilon_t$ is a white noise innovation term drawn from a Gaussian distribution with mean zero and [..[114] ]standard deviation $\sigma$[..[115] ].

[..[116] ]In this study, the LOWESS smoothing window length is 50 years with weights decaying with increasing distance according to a tricube weight function. The regression coefficients for the forced response to volcanic eruptions are obtained

195 with the ordinary least squares (OLS) algorithm. The coefficients of the AR process are fit by means of maximum likelihood and the Bayesian Information Criterion (BIC) [..[117] ]is employed to select [..[118] ]its order p with the maximum [..[119] ]considered order being eight.

**4.1.2 Local [..[120] ]temperature [..[121] ][..[122] ]response module**

The local temperature response module translates the global mean temperature signal into a grid-point level response $T_{s,t}^{resp}$.

200 [..[123] ]Motivated by the pronounced linear scaling of regional land temperatures with global mean temperature (Seneviratne et al., 2016; Wartenburger et al., 2017), the local [..[124] ]response is expressed as

$$T_{s,t}^{resp} = f(T[..[125]]_t^{glob}) = f(\mathsf{T}^{glob,trend}{}_t, T_t^{glob,var}) = \beta[..[126]]^{trend}{}_s \cdot T[..[127]]^{glob,trend}{}_t + \beta_s^{int} + \beta_s^{var} \cdot T_t^{glob,var}[..[128]], \tag{6}$$

with regression coefficients [..[129] ]$\beta_s^{trend}, \beta_s^{int}$, and $\beta_s^{var}$ whereby $\beta_s^{int}$ represents the intercept term. Hence, the response of the local mean temperature to [..[130] ]$T_t^{glob,trend}$ and $T_t^{glob,var}$ are separately taken into account.
* * *
[111]removed: $T_t^{glob,volc}$

[112]removed: $(T_t^{glob} - T_t^{glob,sm})$

[113]removed: unforced global temperature variability $T_t^{glob,var} = T_t^{glob} - T_t^{glob,det}$

[114]removed: its standard deviation set to the empirical standard deviation of the innovations of the training samples (

[115]removed: ).

[116]removed: Here

[117]removed: , which punishes model complexity to avoid overfitting,

[118]removed: the

[119]removed: possible order set to 8.

[120]removed: mean

[121]removed: emulator

[122]removed: The local mean temperature emulator

[123]removed: Due to

[124]removed: mean temperature $T_{s,t}^{resp}$

[129]removed: $\beta_s^{det}, \beta_s^{var}$, and

[130]removed: the deterministic global trend $T_t^{glob,det}$ and the response to the global temperature variability

205    In this study, the linear regression coefficients are estimated with OLS at each grid point.

**4.1.3    Local residual temperature variability [..[131] ]module**

The local residual temperature variability $\eta_{s,t}$ refers to the spatio-temporally correlated residual variability which cannot be accounted for through a response to [..[132] ]$T_t^{glob}$. This variability is assumed to be [..[133] ]Gaussian in nature (see S1 for the results of a Shapiro-Wilks test for normality) and stationary in time which makes it possible to model the time series as local

210    AR(1) processes with spatially correlated innovations (Humphrey and Gudmundsson, 2019). Hence, additional realizations of [..[134] ]$\eta_{s,t}$ are generated stochastically according to

$$\eta_{s,t} = \gamma_{0,s} + [..{}^{135}]\gamma[..{}^{136}]_{1,s} \cdot \eta[..{}^{137}]_{s,t-1} + \nu_{s,t} \quad \text{with} \quad \nu_{s,t} \sim \mathcal{N}(0, \Sigma_\nu(r)), \tag{7}$$

whereby $\gamma_{0,s}$ [..[138] ]and $\gamma_{1,s}$ are the coefficients of the AR model and $\nu_{s,t}$ are [..[139] ]spatially correlated innovations drawn from a multivariate Gaussian with mean zero and [..[140] ]covariance matrix $\Sigma_\nu(r)$ (Cressie and Wikle, 2011).

215    For an AR(1) process, $\Sigma_\nu(r)$ can be analytically derived from the covariance matrix of the residual variability $\Sigma_\eta(r)$ with

$$\Sigma_\nu(r)_{i,j} = \sqrt{1 - \gamma_{1,i}} \cdot \sqrt{1 - \gamma_{1,j}} \cdot \Sigma_\eta(r)_{i,j}, \tag{8}$$

whereby the indices $i$ and $j$ refer to spatial locations $s$ (Cressie and Wikle, 2011).

[..[141] ]To estimate $\Sigma_\eta(r)$, the empirical covariance matrix [..[142] ]$\tilde{\Sigma}_\eta$ is computed. However, $\tilde{\Sigma}_\eta$ is rank deficient because

220    substantially fewer temperature field samples are available than there are land grid points. Thus, $\tilde{\Sigma}_\eta$ needs to be regularized to obtain a robust estimate of the co-variations between the grid points. For this purpose, [..[143] ]we employ localization, an approach which is well established in the field of data assimilation (Carrassi et al., 2018). Localization retains anisotropy on regional scales which is an important asset when stochastically modeling temperature variability since anisotropy is a prevalent feature due to physical factors such as prevailing wind direction and geometry of mountainous terrain. To

225    localize $\tilde{\Sigma}_\eta$, it is point-wise multiplied with a smooth correlation function $G(r)$ with exponentially vanishing correlations with distance:

$$\Sigma_\eta(r) = \tilde{\Sigma}_\eta \circ G(r), \tag{9}$$
* * *
[131] removed: emulator

[132] removed: the global mean temperature signal

[133] removed: stationary in time and

[134] removed: the local residual temperature variability are generated according to

[138] removed: ,...,$\gamma_{p_s,s}$ refer to

[139] removed: the white noise

[140] removed: the regularized empirical spatial covariance matrix $\Sigma(r)$

[141] removed: The rank deficient

[142] removed: $\tilde{\Sigma}$

[143] removed: $\tilde{\Sigma}$ is localized by multiplying it

whereby ∘ denotes the Hadamard product. Here, $G(r)$ is the numerically efficient Gaspari-Cohn function (Gaspari and Cohn, 1999) which vanishes beyond two times the localization radius $L$:

$$
230 \quad G(r) = \begin{cases} 1 - \dfrac{5}{3} \cdot r^2 + \dfrac{5}{8} \cdot r^3 + \dfrac{1}{2} \cdot r^4 - \dfrac{1}{4} \cdot r^5, & \text{if } 0 \leq r < 1, \\ 4 - 5 \cdot r + \dfrac{5}{3} \cdot r^2 + \dfrac{5}{8} \cdot r^3 - \dfrac{1}{2} \cdot r^4 + \dfrac{1}{12} \cdot r^5 - \dfrac{2}{3} \cdot r^{-1}, & \text{if } 1 \leq r < 2, \\ 0, & \text{if } r \geq 2, \end{cases} \tag{10}
$$

with $r = \dfrac{d}{L}$ and $d$ the geographical distance between two grid points.

[..[144] ]

In this study, the AR(1) coefficients are fit at each grid point by means of maximum likelihood. [..[145] ]In our framework implementation, the obtained intercept terms $\gamma_{0,s}$ are effectively zero, as the local response module already contains an intercept term (Eq.6). The localization radius to regularize [..[146] ]$\tilde{\Sigma}_\eta$ is determined by cross-validation [..[147] ]with a leave-one-out approach. Localization radii between 1000 and 4750 km every 250 km [..[148] ]are tested. Thereby, the empirical covariance matrix is estimated based on 230 years and the likelihood to draw the [..[149] ]field of the left-out year from the regularized matrix is computed. [..[150] ]This process is repeated until every year has been left out once for every localization radius. The respective log-likelihood values for each localization radius are summed up [..[151] ]across the left-out years and the radius which is associated with the maximum likelihood is chosen.

**4.2 Evaluating the emulator**

[..[152] ]The emulator's performance is evaluated on the training run and - where available - on test runs. While the evaluation on the training run indicates how successfully this framework implementation captures the training run, the evaluation on the test runs serves as a proxy for the emulator's capability in mimicking true ESM initial-condition ensembles. For the evaluation, 1000 emulations are generated for each climate model.

[..[153] ]
* * *
[144]removed: An ARprocess is

[145]removed: BIC is employed to select the order of the AR processes with the maximum order set to 4. In a subsequent step, the

[146]removed: the empirical covariance matrix of the innovations with

[147]removed: . For this purpose, the time series of 230 years is split into 5 folds. Each fold contains 1 block of 6 years and 8 blocks of 5 years spread out evenly across the ESM run with maximum spacing in time between individual blocks. The empirical covariance matrix of the innovations is estimated based on 4 folds and regularized with localization

[148]removed: . The

[149]removed: innovations of the 5[th] fold from each one of these regularized covariance matrices

[150]removed: The likelihood values

[151]removed: over all folds and finally, the localization radius

[152]removed: To evaluate the emulator, the focus is set on ensemble reliability, i. e., the ability to capture the distribution of ESM runs with an ensemble of emulations (Weigel, 2012). This approach is chosen because the generated emulations cannot be directly compared to ESM runsas they differ in each realization and thus the traditionally employed measure of"best-fit", i.e. , least deviation from the ESM run, is not a meaningful metric.

[153]removed: Visual

**4.2.1 Local trends verification**

The local trends $T_{s,t}^{trend}$ are shared by all emulations and serve as an estimate of the externally forced response with

$$\mathsf{T}_{\mathsf{s,t}}^{\mathsf{trend}} = \beta_{\mathsf{s}}^{\mathsf{trend}} \cdot \mathsf{T}_{\mathsf{t}}^{\mathsf{glob,trend}} + \beta_{\mathsf{s}}^{\mathsf{int}}. \tag{11}$$

[revised manuscript text omitted]

[162]removed: out-of-sample verification on ESM runs not seen during training

[163]removed: the 12 CMIP5 models with several initial-condition ensemble members. In the latter case, the differences between the emulated and counted quantilesare computed individually for each ESM initial-condition ensemble member not seen during training and are subsequently averaged.

[164]removed: selected

[176]removed: calibrated

[177]removed: an emulator on each one of the four selected

[178]removed: deterministic

[179]removed: $T_t^{glob,det}$ diverge by about 1

[180]removed: the global temperature variability

[181]removed: p

[182]removed: between them

[183]removed: mean

[184]removed: $\beta_s^{det}$

[185]removed: spatial patterns of $\beta_s^{det}$

[186]removed: warms less

[187]removed: .

[Figure]

**Figure 3.** Emulator calibration parameters (rows) for [..165 ]four [..166 ]example ESMs (columns). (a) For the global mean temperature [..167 ]module [..168 ]$T_t^{glob,trend}$ and the AR coefficients plus the standard deviation of the innovations of [..169 ]$T_t^{glob,var}$ are depicted. (b) For the local [..170 ]temperature response module, the regression coefficients are shown. (c) For the local residual temperature variability [..171 ]module, the lag-1 AR [..172 ]coefficients, the standard [..173 ]deviations of the innovations[..174 ], and the localization [..175 ]radii are displayed.

[revised manuscript text omitted]

[343]removed: The advantages of a modular framework

[344]removed: The advantages of the modular framework approach chosen in this study are manifold. First, the calibrated parameters of each emulator module serve as an interesting approach for climate model inter-comparison over a wide range of scales which can be clearly visualized and easily interpreted (Sects. 5.1 and 6.1). Second, the modular framework renders it straightforward to substitute each emulator module with approaches other than the ones chosen here. For example, with alternative approaches for the deterministic global mean temperature trend (e.g., Meinshausen et al., 2011), for the local mean temperature model (e.g., Tebaldi and Arblaster, 2014; Alexeeff et al., 2018), or for the local residual temperature variability (e.g., Link et al., 2019). Third, if the modeling task were to change, additional predictors could easily be integrated. For example, precipitation emulation would likely require human-induced aerosol emissions as an additional predictor in the local mean response model (Frieler et al., 2012).

Taylor et al., 2012; Eyring et al., 2016). While multi-model ensembles are imperfect, with several ESMs [..³⁴⁵ ]exhibiting
dependencies (Knutti, 2010; Bishop and Abramowitz, 2013; Sanderson et al., 2015; Abramowitz et al., 2019), multi-model un-
certainty nevertheless clearly exceeds single-model initial-condition uncertainty at the yearly scale for temperature (Sect. 5.3).
ESMs contained within CMIP5 differ [..³⁴⁶ ]substantially across a broad range of scales [..³⁴⁷ ]and thus sample different phase
spaces in projections [..³⁴⁸ ]which renders it necessary to train an emulator on each climate model to approximate the CMIP5
ensemble. A single-model initial-condition ensemble, on the other hand, can be successfully mimicked on grid-point to re-
gional scales by training on a single [..³⁴⁹ ]ESM run (Sects. 5 and [..³⁵⁰ ]6). While this lies beyond the scope of this study, the
developed emulator could additionally serve as a novel tool to address the challenge of inter-model dependencies. Differences
between climate models could be quantified in terms of their [..³⁵¹ ]emulator parameters and subsequently, a subset of models
with sufficiently divergent parameters could be selected to base projections on. Additionally, observations could be used to
constrain the emulated ensemble by providing validation measures for the emulator parameters.

**8  Conclusions and outlook**

We [..³⁵² ]introduce a modular framework for climate model emulation of yearly land temperatures and [..³⁵³ ]present a
specific, computationally cheap implementation, which [..³⁵⁴ ]can create plausible temperature field time series within seconds
based on a single climate model training run. Our emulator consists of (i) a global mean temperature module, (ii) a local
[..³⁵⁵ ]temperature response module, and (iii) a local residual temperature variability module. The global mean temperature
module contains a [..³⁵⁶ ]global mean temperature trend which is shared by all emulations and a global mean temperature
variability term which is modeled as an AR process and varies between individual emulations. The local [..³⁵⁷ ]response
module is linear in nature and consists of a separate response to the [..³⁵⁸ ]global mean temperature trend and the global
mean temperature variability. The local residual variability module generates spatio-temporally correlated fields by means of
locally fit AR(1) processes with spatially correlated innovations.

[..³⁵⁹ ]Since emulators approximate complex ESMs in a simplified manner, they are not able to accurately reproduce all
spatio-temporal ESM characteristics. The emulator presented here, e.g., dampens co-variations between grid points as
* * *
³⁴⁵removed: showing
³⁴⁶removed: sufficiently
³⁴⁷removed: to
³⁴⁸removed: ,
³⁴⁹removed: available
³⁵⁰removed: **??**
³⁵¹removed: calibrated
³⁵²removed: introduced
³⁵³removed: presented
³⁵⁴removed: makes it possible to
³⁵⁵removed: mean temperature
³⁵⁶removed: deterministic trend as well as a stochastic
³⁵⁷removed: mean temperature model
³⁵⁸removed: the deterministic global
³⁵⁹removed: With emulators trained on single runs of four selected ESMs, we showed that the emulations

[revised manuscript text omitted]